# Tissue-resident B cells orchestrate macrophage polarisation and function

Ondrej Suchanek [1,2], John R. Ferdinand[1], Zewen K. Tuong [1], Sathi Wijeyesinghe [3], Anita Chandra [4], Ann-Katrin Clauder [5], Larissa N. Almeida [5], Simon Clare[6], Katherine Harcourt[6], Christopher J. Ward[1], Rachael Bashford-Rogers [7], Trevor Lawley[6], Rudolf A. Manz [5], Klaus Okkenhaug [4], David Masopust [3] & Menna R. Clatworthy [1,2,6] ✉

B cells play a central role in humoral immunity but also have antibody-independent functions. Studies to date have focused on B cells in blood and secondary lymphoid organs but whether B cells reside in non-lymphoid organs (NLO) in homeostasis is unknown. Here we identify, using intravenous labeling and parabiosis, a bona-fide tissue-resident B cell population in lung, liver, kidney and urinary bladder, a substantial proportion of which are B-1a cells. Tissue-resident B cells are present in neonatal tissues and also in germ-free mice NLOs, albeit in lower numbers than in specific pathogen-free mice and following co-housing with 'pet-store' mice. They spatially co-localise with macrophages and regulate their polarization and function, promoting an anti-inflammatory phenotype, in-part via interleukin-10 production, with effects on bacterial clearance during urinary tract infection. Thus, our data reveal a critical role for tissue-resident B cells in determining the homeostatic 'inflammatory set-point' of myeloid cells, with important consequences for tissue immunity.

Over the last decade, there has been a growing appreciation that substantial populations of innate and adaptive immune cells permanently reside in non-lymphoid organs (NLO)[1–3] where they contribute to organ homeostasis, defense, and in some cases, immunopathology[4–6]. Tissue immune responses require the coordinated interaction of these resident immune cell populations, recruitment of their circulating counterparts and communication with neighbouring non-immune epithelial, endothelial and stromal cells, including via cytokine and chemokine production[7–10]. To date, studies of tissue-resident immune cells in NLOs have focused mostly on macrophages and memory CD8+ T cells with limited data on B cells[2,11–14].

B cells, in addition to antibody generation, play important roles in T cell activation[15], pro-inflammatory cytokine production[16–18] and

immune regulation via interleukin-10 (IL10) secretion[19–21]. There are two major subsets of B lymphocytes that originate from different progenitors and have previously been described to have distinct functions and locations; conventional B-2 cells that arise from bone marrow precursors, and an innate-like B-1 subset that arises in embryogenesis from yolk-sac or fetal liver precursors[22,23]. Studies in mice show that B-2 cells dominate in spleen, lymph nodes and blood, whilst B-1 cells are enriched within the pleural and peritoneal cavity, with smaller numbers observed in the spleen. B-1 cells produce IgM natural antibodies and in mice two subsets (B-1a and B-1b) are delineated based on CD5 expression[23]. Although B-1 cells occupy body cavities in homeostasis, their presence in skin, lung and kidney has been described in the context of injury or infection[11,24–28]. As described

[1]Molecular Immunity Unit, University of Cambridge Department of Medicine, Cambridge, UK. [2]Cambridge University Hospitals NHS Foundation Trust, and NIHR Cambridge Biomedical Research Centre, Cambridge, UK. [3]Department of Microbiology and Immunology, Centre for Immunology, University of Minnesota, Minneapolis, MI, USA. [4]Department of Pathology, University of Cambridge, Cambridge, UK. [5]Institute for Systemic Inflammation Research, University of Luebeck, Luebeck, Germany. [6]Wellcome Sanger Institute, Wellcome Genome Campus, Hinxton, UK. [7]Wellcome Trust Centre for Human Genetics, University of Oxford, Oxford, UK. ✉e-mail: mrc38@cam.ac.uk

for tissue-resident memory T cells[13,29], the concept of protective, antigen-specific memory B2 or plasma cells seeding into an NLO following local infection has been recently explored by several studies[14,30–32]. However, whether B cells reside in unchallenged NLOs and contribute to tissue homeostasis and early defense against infection remains unknown. Here we sought to address these questions.

We find that at steady state all major organs examined (lungs, liver, kidney and bladder) house a population of bona-fide tissue-resident B cells, with B-1 cells making up a substantial proportion of this tissue pool, which expanded further in animals with greater microbial exposure. Using the renal tract as an exemplar tissue, we observe the number of tissue-resident B cells inversely correlate with susceptibility to infection following bacterial challenge, suggesting that these B cells may inhibit anti-microbial responses. Indeed, we observe that extravascular B cells have a profound effect on macrophage polarization, promoting an anti-inflammatory phenotype in both tissue-resident and monocyte-derived macrophages, at least in part, via B-cell derived IL10. Together our data suggest a new paradigm for B cell biology: We propose that the homeostatic seeding of B-1 cells is not limited to body cavities and the spleen, but extends to all major organs, in an analogous way to macrophages. These two cell types reside side by side in tissue niches, enabling B cells to shape macrophage polarisation and to set their 'inflammatory set-point', with important consequences for tissue immunity and defence.

## Results

### Extravascular B cells present across non-lymphoid murine organs in homeostasis

In order to examine extravascular immune cells in murine organs we administered an anti-CD45 antibody to specific pathogen-free (SPF) C57BL/6 mice intravenously pre-mortem, as described previously[33]. We profiled a variety of NLOs (kidneys, urinary bladder, lungs and liver) that differ in the extent to which they interface with the external environment, as well as spleen, blood and peritoneal cavity. This revealed differing numbers of extravascular B cells (CD45$^+$CD19$^+$) across NLOs, with the highest number in the liver (Fig. 1a, b, Supplementary Fig. 1a). B cells also formed a variable proportion of the total extravascular immune compartment, comprising 5–10% of all CD45$^+$ kidney or lung immune cells, and up to 20% of liver immune cells (Fig. 1c). When examining the phenotype of extravascular B cells relative to those in blood, we observed a reduction in the proportion of naïve (IgM$^+$IgD$^+$) cells and a marked increase in IgM$^+$IgD$^-$ and to lesser extent IgM$^-$IgD$^-$ (double negative, DN) B cells in kidney, bladder and lung but not in liver (Fig. 1a, d). Similarly, in human kidneys, we found an enrichment of IgD$^-$ B cells compared with spleen (Fig. 1e, Supplementary Fig. 1b).

To further characterise IgD$^-$ B cells enriched within murine organs, we examined their expression of markers associated with B-1 cells and IL10-producing regulatory B cells (CD9)[34] (Fig. 1a, Supplementary Fig. 1c). We found that the extravascular B cell pool was substantially enriched for B-1a cells (B220$^{low}$CD23$^-$CD21$^{low}$IgD$^{low}$IgM$^{hi}$CD5$^+$) and CD9$^+$ cells across kidney, bladder, lung and liver compared with blood and spleen, in some organs with similar proportions to that observed in the peritoneal cavity (Fig. 1f, Supplementary Fig. 1c). However, further immunophenotyping revealed that these two populations largely overlap as extravascular B-1a cells across NLOs also expressed CD9 (Supplementary Fig. 1c). Again, the liver differed from the other organs profiled with a smaller proportion of extravascular B-1a cells compared to kidney, bladder, and lung (<25% versus 40%, 60% and 40%, respectively) (Fig. 1f). B-1 cells have been described to seed body cavities pre-natally[22,23]. To test whether CD5$^+$ B-1a cells become established in other NLOs, in an analogous way to macrophages, we examined organs obtained from neonatal mice, demonstrating the presence of CD5$^+$ B cells extravascularly (Fig. 1g, Supplementary Fig. 1d).

### Bona-fide tissue-resident B cell compartments in non-lymphoid organs confirmed by parabiosis

To explore whether extravascular B cells in NLOs were bona-fide tissue-resident cells, we performed a parabiosis experiment between CD45.1 and CD45.2 mice. Five weeks after parabiosis was established there were equal proportions of CD45.1 and CD45.2 B cells in peripheral blood (Fig. 2a). In contrast, within the extravascular compartment of kidney, liver, lung and fat, a significantly higher proportion (70–80%) of IgM$^+$IgD$^-$ and B-1a cells remained of host origin than that observed for naïve B cells (50–60%) (Fig. 2b, Supplementary Fig. 2a), suggesting that these two subsets are particularly enriched in the homeostatic tissue resident pool and have limited exchange with circulating B cells. This mirrors the findings by Ansel et al. who demonstrated lower mixing of B-1a cells compared to B-2 cells in peritoneal cavity after 8-week parabiosis[27].

Tissue-resident populations have the capacity for self-renewal[35]. To test whether extra-vascular B cells could proliferate in situ to re-establish tissue populations following depletion, we administered a murine anti-CD20 antibody intravenously, followed by EdU in drinking water for 3 weeks (Fig. 2c). Interestingly, the extent to which extravascular B cells were depleted following antibody administration varied across organs, and was almost complete in kidney and liver, whilst bladder and lung B cells were resistant to depletion (Fig. 2d), suggesting that B cells in kidney and liver may be anatomically located within sub-niches that are more accessible to antibody diffusing into the tissue from the circulation. Notably these organs contain fenestrated capillaries (kidney) or sinusoids (liver, spleen). Alternatively, it may reflect a differential susceptibility to antibody-dependent cellular cytotoxicity of B cells in these organs. Overall, naïve extravascular B cells in NLOs were more susceptible to depletion than IgM$^+$IgD$^-$ or DN subsets (Supplementary Fig. 2b, c). Following depletion, EdU incorporation was at least twice as frequent in these two IgD$^-$ subsets (including B-1a cells) as that present in naïve B-2 cells (Fig. 2e, Supplementary Fig. 2d), confirming their capacity to proliferate in-situ.

### Strain-specific and microbiome-dependent effects on tissue-resident B cells

We next considered factors that might influence the nature and magnitude of the tissue-resident B cell compartment. Different inbred strains of mice are known to have differences in baseline polarization of immune responses, for example, C57BL/6 and BALB/c mice have Th1 and Th2 polarised responses, respectively[36–38]. To test the extent to which genetic background of different inbred strains might impact the tissue-resident B cell compartment, we examined extravascular B cells in C57BL/6, BALB/c and DBA mice bred at the same animal facility. This demonstrated significant strain-related differences, with DBA mice showing a lower number of extravascular B cells in all organs examined (Fig. 3a). The proportion of B-1a cells also differed between strains, with BALB/c mice showing the highest proportions in NLOs and the peritoneal cavity (Fig. 3a).

The microbiome has a substantial influence on tissue immunity, particularly in organs that interface with the environment[12,39–42]. In germ-free (GF) mice, extravascular B cells, including B-1a cells, were present in NLOs, but were lower in number compared with SPF mice at the same facility, and an independent SPF animal facility with a lower pathogen barrier level (Fig. 3b). Of note, SPF mice had a significantly higher extravascular B cell count and percentage of B-1a cells within NLOs when housed in the animal facility with a lower pathogen barrier level. (Fig. 3b). To explore this further, we co-housed SPF mice with pet store mice which have a more diverse microbiome than SPF animals[43]. After 60 days of co-housing, we observed higher numbers of extravascular B cells in both an environment-facing organ, namely the lung and a sterile organ that does not directly contact the exterior, the kidney, in co-housed SPF mice compared to their non-cohoused SPF controls (Fig. 3c). The proportion of non-naïve B cells was also

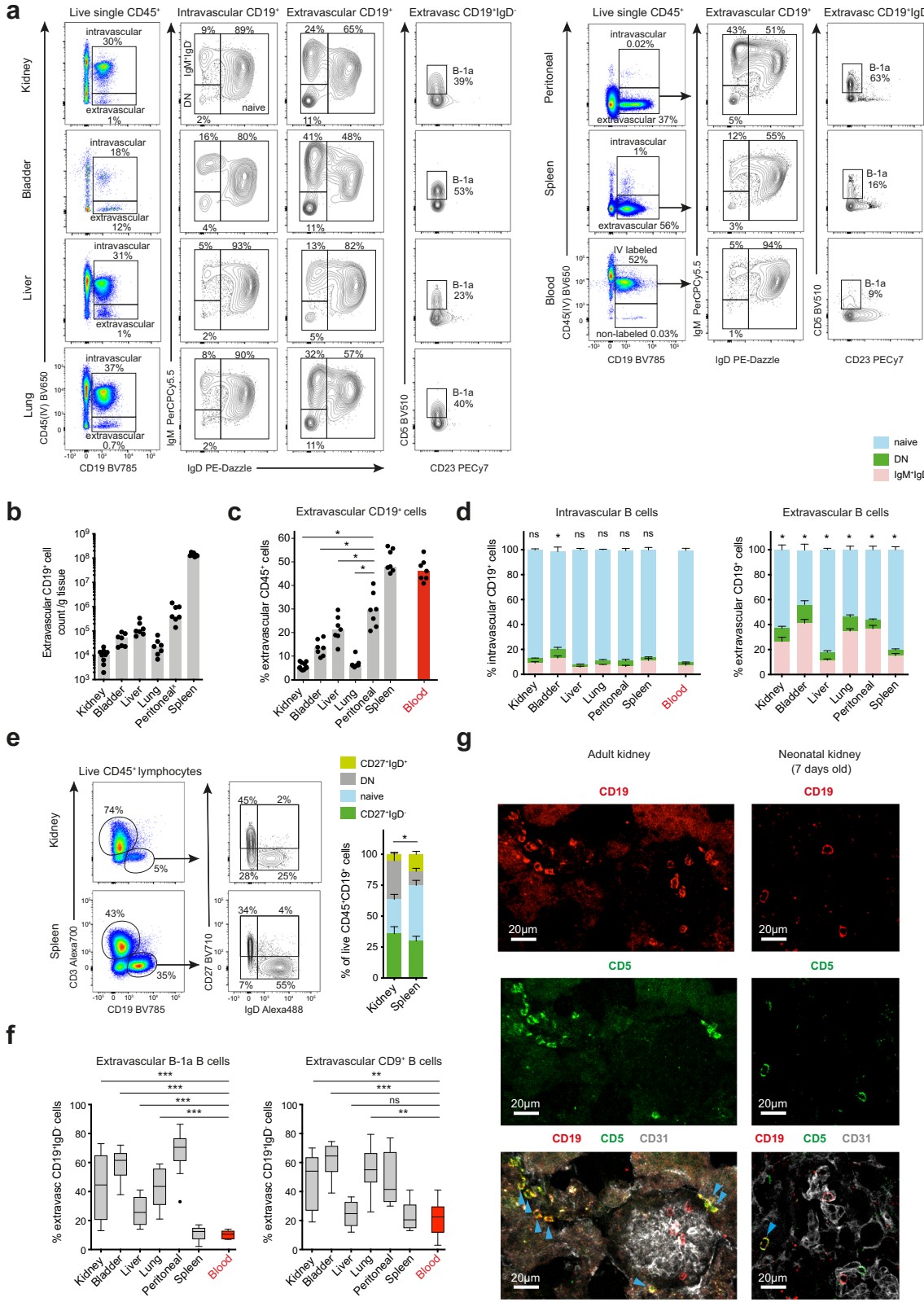

increased in co-housed mice compared with their non-cohoused counterparts, particularly in the lung (Fig. 3d). Together these data demonstrate that the microbiome may either directly or indirectly shape the magnitude and phenotype of the B cell compartment across NLOs, including tissues that do not directly interface with the environment. Interestingly, extravascular B cells, including B-1a cells, in the kidney and bladder increased with age (Fig. 3e), which mirrors

previous reports of age-related B-1 cell expansion in the peritoneal cavity[44,45].

**Extravascular B cell repertoire in the renal tract is less diverse and is expanded following bacterial challenge**

Increased numbers of B cells have been noted in the lungs in bacterial and viral infections[14,26,31,46]. We therefore examined extravascular B

**Fig. 1 | Extravascular B cells present across murine organs in homeostasis.**
**a** Flow-cytometry profiling of extra- and intravascular CD19⁺ cells in WT C57BL/6 mouse kidney, urinary bladder, liver, lung, spleen, peritoneal lavage and blood (kidneys $n = 9$, other organs $n = 7$). Data representative of three independent experiments. **b** Absolute extravascular B cell counts per gram of tissue across organs shown in **a**. ˣPeritoneal B cell counts calculated per single lavage and adjusted for total volume of PBS injected. **c** Percentage of B cells within live extravascular CD45⁺ cell subset across organs shown in **a**. Kidney *$P = 0.0156$, bladder *$P = 0.0156$, liver *$P = 0.0312$, lung *$P = 0.0156$. **d** Proportion of naïve (IgD⁺IgM⁺), IgM⁺IgD⁻ and switched (double negative, DN) B cells from intra- (left) and extravascular (right) compartments in organs shown in **a**, $n = 7$ per group, means with SEM shown. The difference in naïve B cells proportions between blood and a tested organ is compared; intravascular: kidney $P = 0.2188$, bladder *$P = 0.0156$, liver $P = 0.1562$, lung $P = 0.3750$, peritoneal $P = 0.6250$, spleen $P = 0.0781$; extravascular: *$P = 0.0156$. **e** Flow-cytometry identification of CD45⁺CD3⁻CD19⁺ cells in matched kidney and spleen from deceased human organ donors (left) and their phenotyping based on expression of IgD and CD27 (middle).

Percentage of naïve (IgD⁺CD27⁻), switched memory (IgD⁻CD27⁺), DN and double positive cells within CD45⁺CD3⁻CD19⁺ cell subset shown (right) ($n = 10$ per group, means with SEM shown). The difference in naïve B cells proportions between kidney and spleen is tested, *$P = 0.0137$. **f** Tukey plots show percentage of B-1a (CD19⁺CD23⁻CD5⁺) and CD9⁺ cells in extravascular IgD⁻ B cell compartments of organs described in **a**. Data pooled from three independent experiments at two independent SPF animal facilities (total $n = 12$ per organ); %B-1: kidney ***$P = 0.0010$, bladder, liver and lung ***$P = 0.0005$; %CD9⁺ cells: kidney **$P = 0.0024$, bladder ***$P = 0.0005$, liver $P = 0.4785$, lung **$P = 0.0029$. **g** Confocal microscopy of adult (left) and neonatal (right) renal cortex from WT C57BL/6 mice focusing on extravascular localization of CD5⁺ B cells. Sections stained for CD19 (red), CD5 (green) and CD31 (grey). Representative of three independent experiments. Blue arrows point at extravascular CD19⁺CD5⁺ B cells. Blood samples represent always only intravascular cells. $P$ values were calculated using two-tailed Wilcoxon matched-pairs signed rank test (c-f). Bar plots show medians unless stated otherwise. Source data are provided with this paper.

cells in the bladder and kidney following challenge with uropathogenic *Escherichia coli* (UPEC), a clinically relevant pathogen of the renal tract (Fig. Supplementary Fig. 3a). At early time-points following infection, bacterial colonies were detectable in both bladder and kidney (Supplementary Fig. 3b), and there was recruitment of neutrophils and monocytes (Fig. Supplementary Fig. 3c, d), in line with previous reports[47,48]. There was no statistically significant change in extravascular B cell numbers or proliferation following UPEC challenge, but evidence of increased activation of some subsets, as indicated by expression of MHCII (Supplementary Fig. 3e, f).

To assess whether bacterial challenge might influence the BCR repertoire, we FACS sorted B cells from the intra- and extravascular B cell compartment of mouse kidneys in homeostasis and 56 days following UPEC challenge, and performed BCR sequencing (Fig. 4a, Supplementary Fig. 3g). In unchallenged kidneys, we found a significantly higher usage of *Ighm* (Supplementary Fig. 3h) and $V_H$-11 and $V_H$-12 segments, less diverse BCR repertoire with smaller sized random non-templated nucleotide additions in the extravascular compartment, consistent with the presence of prenatally generated B-1 cells[49,50] (Fig. 4b–e). The extravascular compartment had a higher frequency of 'public' BCR clonotypes, although these represented less than 10% of total reads (Fig. 4f, g). There was also a number of expanded, dominant clones within the extravascular B cell compartment, the CDR3 sequence of which was found in the intravascular compartment (Fig. 4h). Thus, this BCR analysis supports the conclusion that some B cell clones enter the kidney from the blood and subsequently undergo local activation and somatic hypermutation (Fig. 4i, j). Following UPEC challenge, the extravascular BCR diversity and unique clonotypes number became more similar to the intravascular compartment (Fig. 4e, k, l), suggesting that infection promotes the entry of intravascular (mostly naïve) B cells with new BCR specificities to the kidney. This was supported by the observed reduction in the number of $V_H$-segment gene mutations observed in the extravascular kidney B cell compartment following infection (Supplementary Fig. 3i).

### Tissue-resident B cells impact bacterial defence in the renal tract

We next identified two strains of mice with higher or lower numbers of extravascular B cells, particularly B-1a cells, to test the function of tissue-resident B cells in the context of an early local infection; we used µMT⁻ mice that are profoundly deficient in B-2 cells and have very low numbers of B-1a cells within organs, and p110δ^E1020K-B mice that have significantly higher numbers of B-1a cells in NLOs, peritoneal cavity and spleen because of a B-cell specific activating mutation in the catalytic subunit of phosphoinositide-3-kinase δ driven by *Mb1*^cre[46,51] (Fig. 5a, Supplementary Fig. 4a). Surprisingly, at early time points (6-12 hrs) following UPEC challenge, we observed reduced

numbers of colony forming units (CFU) in the bladders and kidneys of B cell deficient µMT⁻ mice compared with WT counterparts. Remarkably, most µMT⁻ kidneys were completely sterile (Fig. 5b, Supplementary Fig. 4b) and there was increased recruitment of neutrophils and inflammatory monocytes to the renal tract of these mice (Fig. 5c). Neutrophil and monocyte-recruiting chemokine transcripts were also more highly expressed within µMT⁻ kidneys compared to controls (Fig. 5d, Supplementary Fig. 4c). In contrast, p110δ^E1020K-B mice demonstrated a greater bacterial burden than controls (Fig. 5e), with reduced neutrophil and monocyte recruitment (Fig. 5f) and lower transcripts of chemokines capable of recruiting these cells (Fig. 5g, Supplementary Fig. 4d). These data suggest that tissue-resident B cells negatively may regulate anti-bacterial responses in the renal tract, impacting the recruitment of circulating myeloid phagocytes to infected organs.

### Tissue-resident B cells orchestrate macrophage polarisation in the renal tract

Mononuclear monocytes (MNP) are the most abundant tissue-resident immune cell within the urinary tract and play an essential role in early responses to bacterial infection[10,48,52]. To explore whether tissue-resident B cells might influence the homeostatic recruitment and function of tissue MNPs, we sought to more fully characterize tissue MNPs within the bladder and kidney and to determine the effect of B cells on this compartment. Analysis of single-cell transcriptomes from wild-type mouse kidney[53] revealed two major macrophage subsets (Supplementary Figs. 5a, b), and identified distinct surface markers that could be utilized for flow cytometric assessment of Ly6C/G- cells: an F4/80^high CD11b^low population (referred to here as Mac1) and an F4/80⁺CD11b^high population (referred to here as Mac2) (Supplementary Figs. 5c, d). Interestingly, these markers have previously been used to distinguish fate-mapped yolk-sac and monocyte-derived macrophages respectively[54], although recent studies suggest that no surface markers infallibly identify ontogeny[55]. In B cell deficient µMT⁻ mice, we observed an increase in Mac2 absolute count compared to their WT counterparts (Fig. 6a), even at steady state, in the absence of any challenge. Of note, experimentally, macrophages may be polarized towards extremes of a pro-inflammatory phenotype, classically by stimulation with lipopolysaccharide (LPS) or an anti-inflammatory phenotype by incubation with IL4[56], the latter expressing CD206. In the renal tract, the phenotype of the Mac2 population differed between µMT⁻ and WT mice, with a lower proportion of CD206-expressing cells in µMT⁻ mice (Fig. 6b, Supplementary Fig. 5e). Conversely, in p110δ^E1020K-B mice we observed a lower number of Mac2 cells (Fig. 6c) and higher proportion of CD206⁺ Mac2 cells in the bladder, although not in the kidney (Fig. 6b, Supplementary Fig. 5e). Furthermore, when considering the relationship between tissue MNP subsets and resident

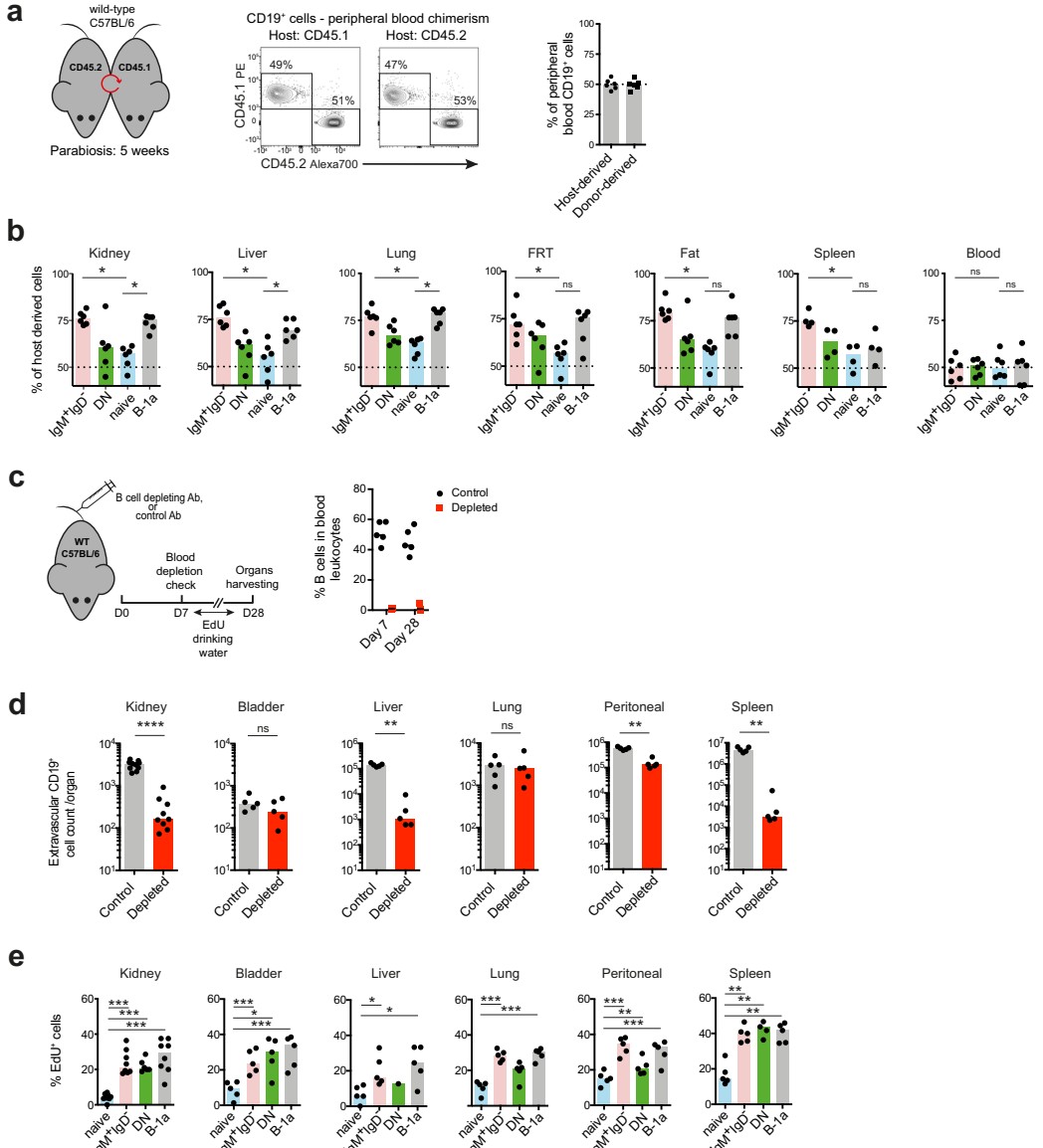

**Fig. 2 | Bona-fide tissue-resident B cell compartments in non-lymphoid organs confirmed by parabiosis. a** Schematic showing parabiosis experimental setup (left): Congenically marked WT animals were connected for 5 weeks before organs harvesting. Flow cytometry (middle) and quantification (right) of peripheral blood CD19+ lymphocytes chimerism in each parabiont, (n = 6 per group). Data representative of two independent experiments. **b** Percentages of host-derived extravascular IgM+IgD-, DN, naïve and B-1a B cells found in kidney, liver, lung, female reproductive tract (FRT), spleen and blood from parabiosis experiment described in **a**. P values were calculated with two-tailed Wilcoxon matched-pairs signed rank test: *P = 0.0312, FRT & Fat (B-1a) P = 0.0625, spleen (B-1a) P = 0.1250, blood (IgM+IgD-) P = 0.8438, blood (B-1a) P = 0.9999. **c** Schematic showing B-cell depletion experimental setup (left): WT C57BL/6 mice were injected with anti-mouse CD20 depleting antibody or isotype control. On day 7, EdU was introduced into their drinking water for 21 days, after which organs (incl. blood) were harvested. Percentage of B cells in peripheral blood leukocytes from these mice was checked both on day 7 and 28 (right, n = 5 per group). Data representative of two

independent experiments. **d** Absolute extravascular CD19+ cell counts found in kidney (controls n = 10, depleted n = 9, ****P < 0.0001), urinary bladder (n = 5, P = 0.5476), liver (n = 5, **P = 0.0079), lung (n = 5, P = 0.6905), peritoneal wash (n = 5, **P = 0.0079) and spleen (n = 5, **P = 0.0079) from depleted mice or controls on day 28 as described in **c**. P values were calculated with two-tailed Mann-Whitney U test. **e** Percentage of EdU positive cells in extravascular naïve, IgM+IgD-, DN and B-1a B cell compartment found in organs from circulating B-cell depleted mouse cohort as described in **c**. P values were calculated with paired two-tailed t test (data normality confirmed with Shapiro-Wilk test): Kidney (n = 8) ***P = 0.0003 (IgM+IgD-), ***P = 0.0002 (DN), ***P = 0.0004 (B-1a); bladder (n = 5) ***P = 0.0005 (IgM+IgD-), *P = 0.0313 (DN), ***P = 0.0010 (B-1a); liver (n = 5) *P = 0.0193 (IgM+IgD-), *P = 0.0119 (B-1a); lung (n = 5) ***P = 0.0006 (IgM+IgD-), ***P = 0.0004 (B-1a); peritoneal (n = 5) ***P = 0.0001 (IgM+IgD-), **P = 0.0060 (DN), ***P = 0.0006 (B-1a); spleen (n = 5) **P = 0.0047 (IgM+IgD-), **P = 0.0089 (DN), **P = 0.0055 (B-1a). Representative flow cytometry plots are shown in Supplementary Fig. 2d. All box plots show medians. Source data are provided with this paper.

B cells across liver, lung, kidney and bladder, there was a strong positive correlation between the %CD206+ Mac2 cells and the percentage of B-1a B cells within organs (Fig. 6d). This suggests that the phenotype of monocyte-derived macrophages in the renal tract is shaped by tissue-resident B cells, which promote anti-inflammatory polarisation. Indeed, B cells within the kidney were observed in close

proximity with MNPs (Fig. 6e, Supplementary Fig. 5f), readily enabling cellular cross-talk.

To perform a more robust assessment of tissue macrophage polarisation changes in the absence or presence of tissue-resident B cells, we flow-sorted Mac1 and Mac2 populations from murine kidney cell suspensions obtained from WT and µMT- mice following a 2-hour

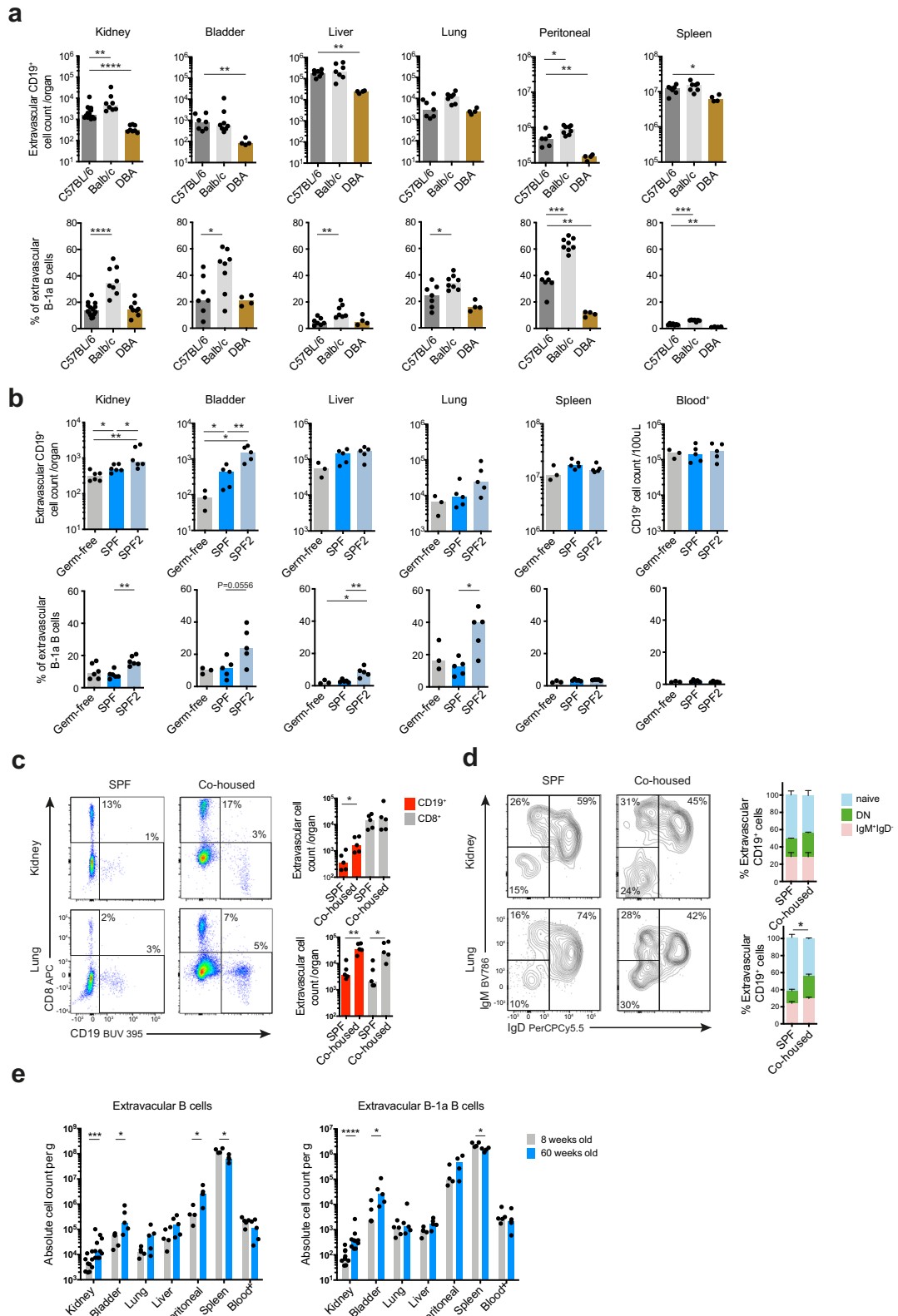

stimulation with lipopolysaccharide (LPS), and performed RNA sequencing on sorted subsets. This demonstrated substantial transcriptional differences in both macrophage subsets between the two strains (Supplementary Fig. 6a–c). Gene set enrichment analysis showed that both Mac1 and Mac2 cells from μMT⁻ mice had increased expression of 'interferon gamma response' and 'interferon alpha response' pathway genes, as well as other innate immune pathways

(Fig. 6f). Furthermore, alignment of our kidney macrophages datasets with the transcriptional profiles of a spectrum of human macrophage activation states[57] revealed enrichment of genes induced by pro-inflammatory stimuli, including LPS and interferon (IFN)-γ in μMT⁻ kidney MNPs (Supplementary Fig. 6d). In p110δ[E1020K-B] mice, kidney Mac1 were enriched for oxidative phosphorylation genes (Fig. 6f), the preferred metabolic pathway used in anti-inflammatory

**Fig. 3 | Strain-specific and microbiome-dependent effects on tissue-resident B cells. a** Absolute extravascular CD19$^+$ cell counts (top) and percentage of B-1a B cells within the extravascular B cells (bottom) found in organs from three commonly used WT laboratory mouse strains: C57BL/6, Balb/c and DBA. Data pooled from two independent experiments. Kidney (C57BL/6 $n = 14$, Balb/c $n = 8$, DBA $n = 8$): top $^{**}P = 0.0064$, $^{****}P < 0.0001$, bottom $^{****}P < 0.0001$; bladder (C57BL/6 $n = 7$, Balb/c $n = 8$, DBA $n = 4$): top $^{**}P = 0.0061$, bottom $^*P = 0.0401$; liver (C57BL/6 $n = 7$, Balb/c $n = 7$, DBA $n = 4$): top $^{**}P = 0.0061$, bottom$^{**}P = 0.0041$; lung (C57BL/6 $n = 7$, Balb/c $n = 8$, DBA $n = 4$): bottom $^*P = 0.0289$; peritoneal (C57BL/6 $n = 6$, Balb/c $n = 8$, DBA n $= 4$): top $^*P = 0.0127$, $^{**}P = 0.0095$, bottom $^{***}P = 0.0007$, $^{**}P = 0.0095$; spleen (C57BL/6 $n = 7$, Balb/c $n = 7$, DBA $n = 4$): top $^*P = 0.0242$, bottom $^{***}P = 0.0006$, $^{**}P = 0.0061$. **b** Absolute extravascular CD19$^+$ cell counts (top) and B-1a cell percentage within extravascular B cell compartment (bottom) found in NLOs, spleen and peripheral blood from WT C57BL/6 mice maintained under different barrier status: Germ-free (kidney $n = 6$, other organs $n = 3$), specific-pathogen-free (SPF, kidneys $n = 6$, other organs $n = 5$) and specific-pathogen-free in an animal facility with a lower barrier status (SPF2, kidneys $n = 6$, other organs $n = 5$). Kidney: top $^*P = 0.0260$, $^{**}P = 0.0022$, bottom $^{**}P = 0.0022$; bladder: top $^*P = 0.0357$, $^{**}P = 0.0079$, bottom $P = 0.0556$; liver: bottom $^*P = 0.0357$, $^{**}P = 0.0079$; lung: bottom $^*P = 0.0159$. **c** Flow cytometry plots (left) and absolute counts (right) of live single extravascular CD8$^+$ and CD19$^+$ cells found in kidney and lung from SPF WT C57BL/6 mice co-housed with pet store mice (Co-housed) for 60 days and their non-cohoused matched controls (SPF), $n = 5$ per group, data representative of two independent experiments. Kidney: $^*P = 0.0317$; lung: $^{**}P = 0.0079$, $^*P = 0.0159$. **d** Flow cytometry phenotyping and percentages of naïve (IgD$^+$IgM$^+$), DN and IgM$^+$IgD$^-$ extravascular CD19$^+$ cells from SPF and co-housed mice as described in **c**, $n = 5$ per group, means with SEM shown, $^*P = 0.0286$. The statistical testing comparing the difference in naïve B cells proportions between SPF and co-housed animals. **e** Absolute extravascular B cell (left) and B-1a cell counts per gram of kidney ($n = 10$), urinary bladder ($n = 5$), liver ($n = 5$), lung ($n = 5$), peritoneal wash ($n = 4$) and spleen ($n = 4$) and 100 uL blood ($n = 5$) from 8- and 60-week old WT female C57BL/6 mice. Kidney: left $^{***}P = 0.0004$, right $^{****}P < 0.0001$; bladder & peritoneal: $^*P = 0.0317$; spleen: $^*P = 0.0286$. All $P$ values were calculated with two-tailed Mann-Whitney U test. Box plots show medians unless stated otherwise. Source data are provided with this paper.

macrophages[58]. Furthermore, Mac2 were enriched for genes upregulated in IL4-stimulated macrophages, a classical anti-inflammatory stimulus (Supplementary Fig. 6e).

Analysis of differential gene expression revealed that monocyte-recruiting chemokines *Ccl2* and *Ccl7* were more highly expressed in Mac1 cells obtained from μMT$^-$ kidneys (Fig. 6g), as well as higher levels of *Il1b*, *MhcII* and *Cd86* transcripts in Mac2 (Supplementary Fig. 6b). In p110δ$^{E1020K-B}$ mice, we observed lower expression of *Ccl3* in both Mac1 and Mac2 compared with controls (Fig. 6g), as well as lower *Cxcl2* (a neutrophil recruiting chemokine), *Il1b* and higher *Tgfb* transcripts in p110δ$^{E1020K-B}$ kidney Mac2 (Supplementary Fig. 6c).

Given that the magnitude of the tissue B cell pool influenced the homeostatic polarization of tissue MNPs, we asked if it might also impact MNP phagocytic activity. We quantified phagocytosis of GFP-UPEC by bladder neutrophils and macrophages following in vivo intravesical challenge and found reduced uptake of bacteria by bladder neutrophils and macrophages in p110δ$^{E1020K-B}$ mice (with increased numbers of tissue B cells) (Fig. 6h, Supplementary Fig. 6f). In contrast, the opposite was observed in μMT$^-$ bladders, with increased UPEC phagocytosis in neutrophils and Mac2s in the absence of B cells (Fig. 6i, Supplementary Fig. 6g).

Altogether these data suggest that tissue B cells may have a substantial effect on the basal metabolic and functional state of tissue macrophages, including their ability to phagocytose bacteria and to orchestrate neutrophil and monocyte recruitment, consistent with our enumeration of these cell types in the context of infection (Fig. 5c, f).

IL10 is known to act as an immunoregulatory cytokine and has profound effects on macrophages, acting to oppose the metabolic switch to glycolysis induced by inflammatory stimuli such as LPS, whilst suppressing mammalian target of rapamycin (mTOR) activity and promoting mitophagy[59]. We hypothesized that this may be of relevance to the pro-inflammatory kidney MNPs phenotype that we observed in μMT$^-$ mice, as some B cells subsets have previously been shown to produce IL10, including B-1a cells[60]. Analysis of μMT$^-$ kidneys compared with WT controls demonstrated reduced *Il10* transcripts in μMT$^-$ mice and increased *Il10* transcripts in p110δ$^{E1020K-B}$ mice (Fig. 7a) while extravascular B cells were found an important source of IL10 (Fig. 7b). These IL10-producing B cells were predominantly in the tissue-resident IgM$^+$IgD$^-$/B-1a subset, and compared to wild-type mice, their frequency was significantly increased in p110δ$^{E1020K-B}$ kidneys (Fig. 7c, Supplementary Fig. 7a) and other NLOs (Supplementary Fig. 7b).

To test whether B cell-derived IL10 could specifically alter basal tissue macrophage polarization in homeostasis, we performed single-cell (sc) RNA sequencing of flow-sorted CD45$^+$ kidney cells harvested from B-cell specific IL10 knockout (B-IL10-KO) mice (*Cd19-cre$^{+/-}$Il10$^{fl/fl}$*)

and their controls (*Cd19-cre$^{-/-}$Il10$^{fl/fl}$*) (Fig. 7d). This identified two transcriptionally distinct kidney macrophage clusters, one with high expression of *Adgre1* (F4/80) and the other with low expression of *Adgre1* and high expression of *Itgam* (CD11b), analogous to the Mac1 and Mac2 populations we identified by flow cytometry (Fig. 7e, Supplementary Fig. 8a). In B-IL10-KO kidneys, we observed a significant enrichment of '*interferon alpha response*' and '*interferon gamma response*' Hallmark genesets in both macrophage populations (Fig. 7f, Supplementary Fig. 8b, c), reminiscent of the pro-inflammatory tuning of kidney macrophages we present in B cell-deficient μMT$^-$ mice (Fig. 6f). Furthermore, kidney macrophages from B-IL10-KO mice, particularly the *Adgre1*-high Mac1 cluster, showed higher expression of monocyte-recruiting chemokines (*Ccl2*, *Ccl3* and *Ccl4*) compared to control mice (Fig. 7g). To test whether these transcriptomic changes have also an impact on kidney macrophage function, we performed an ex-vivo phagocytosis assay with fluorescently labelled *E. coli* bioparticles. We found that both macrophage subsets in B-IL10-KO kidneys had higher phagocytic activity compared to controls (Fig. 7h). This supports the thesis that the mechanism by which tissue-resident B cells orchestrate macrophage polarization towards an anti-inflammatory phenotype in the renal tract is, at least in part, via production of IL10.

Finally, to test the functional importance of B-cell derived IL10 on local bacterial clearance in vivo, we performed a UTI (cystitis model) in B-IL10-KO mice. We observed significantly greater UPEC clearance from the bladder in B-IL-10 KO mouse compared to controls (Fig. 7i), accompanied by a higher neutrophil and monocyte recruitment to the tissue (Fig. 7j).

## Discussion

Our work shows that in homeostasis, major NLOs, including lung, liver, kidney and urinary bladder harbor B-1 cells and they formed a large component of the tissue-resident B cell pool in these organs. B-1 cell progenitors emerge pre-natally from yolk-sac or fetal liver to populate body cavities and spleen[22,61]. Our data suggests that this early seeding extends beyond body cavities to many or even all organs, in an analogous way to that described for yolk sac-derived macrophages[54,62]. We found extravascular B cells and MNPs in close physical proximity, enabling cross-talk. Indeed, we show that B cells have a substantial impact on the transcriptome of tissue-resident macrophage subsets, increasing the expression of OXPHOS pathway genes and inhibiting the expression of monocyte-recruiting chemokines. We propose that this homeostatic regulation of tissue MNP phenotype represents a central function for tissue-resident B-1 cells, influencing local tissue defence, in addition to their capacity to produce natural antibodies[63].

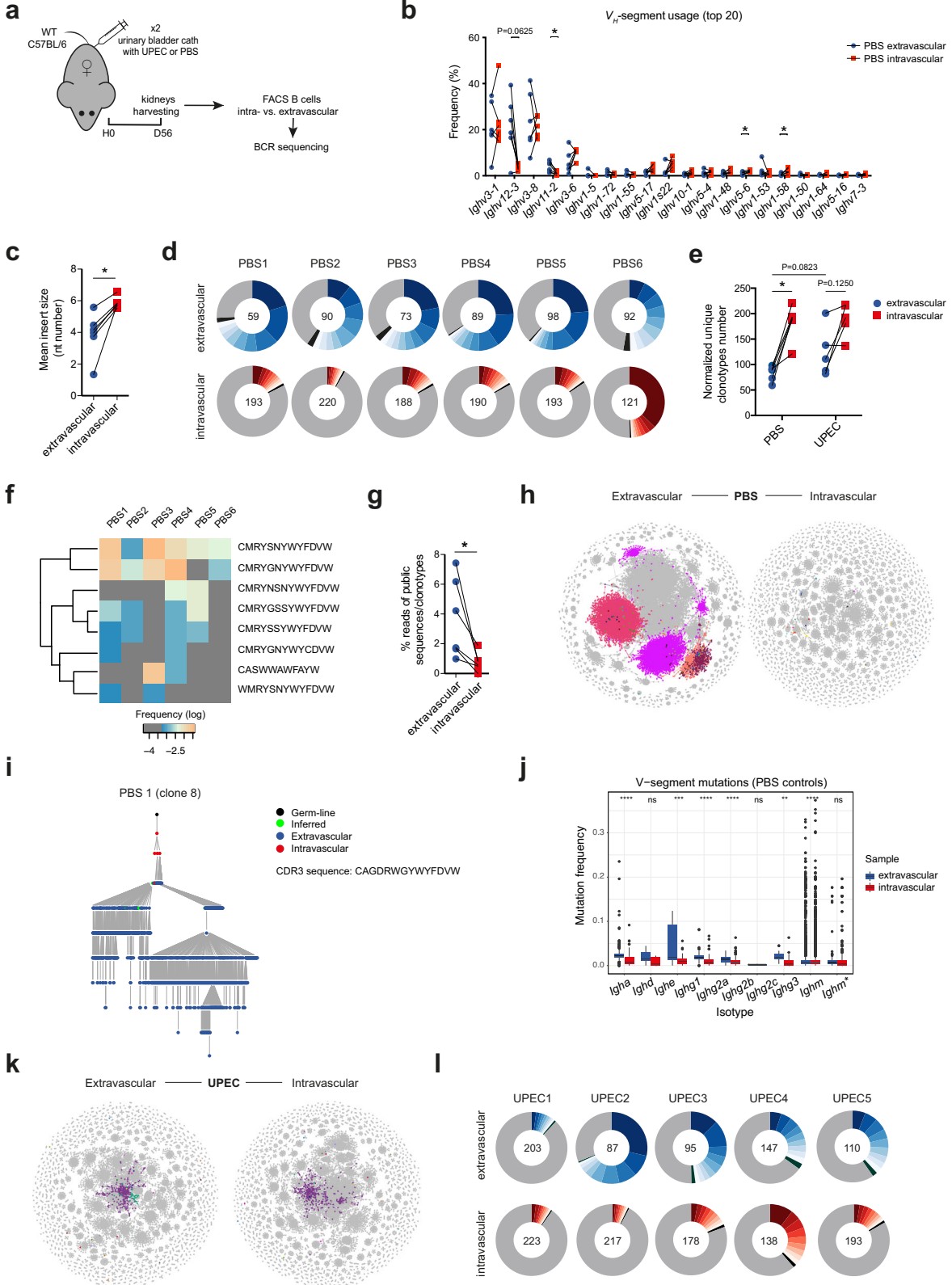

Notably, the size and nature of the tissue-resident B cell compartment differed between organs, with B-1a cells more highly represented in the environment-facing tissues of bladder and lung, compared with kidney and liver. This observation may well be driven by microbial cues, since extravascular B cell compartment was expanded after co-housing with 'pet-store' mice, mirroring studies of tissue-resident memory T cells[43]. However, extravascular B cells were

still present in GF mice, demonstrating that their basal seeding is not dependent on the microbiome, as is the case with neonatally seeded tissue macrophages, but the size of the population post-natally is influenced by the microbiome. This is in keeping with observations that GF mice have normal levels of natural antibodies[64]. The mechanism by which the microbiome influences the tissue B cell compartment may occur via microbial metabolites, as has been described for

**Fig. 4 | Tissue-resident B cell repertoire is less diverse and is expanded following bacterial challenge. a** BCR-seq experimental setup: WT C57BL/6 female mice had urinary bladders inoculated twice with UPEC (*n* = 5), or PBS (*n* = 6). After 56 days, their intra- and extravascular kidney B cells were sorted and BCR-sequenced. **b, c** Comparison of 20 most abundant $V_H$-segment frequencies (**b**) and mean insert sizes (**c**) between matched extra- and intravascular kidney BCR compartments from control (PBS) animals, *$P$ = 0.0312. **d, l** Donut plots showing proportion of ten most abundant *Igh* clonotypes (each clone coloured) found in matched extra- and intravascular kidney BCR compartments from control (PBS) (**d**) and infected (UPEC) mice (**l**). Normalized number of unique *Igh* clonotypes found in each sample is indicated in the centre of each donut. **e** Pair-wise BCR repertoire diversity comparison of extra- and intravascular kidney B cells from control (PBS) and infected (UPEC) mice based on normalized number of unique *Igh* clonotypes per sample, *$P$ = 0.0312. **f** Heatmap showing log frequencies of "public" *Igh* clonotypes (complete CDR3, V- and J-segment nucleotide match), shared by extravascular B cells across control (PBS1-6) kidneys. Clonotypes CDR3 amino-acid sequence displayed. **g** Pair-wise comparison of percentage of "public" *Igh* clonotypes (defined in **f**) reads within extra- vs. intravascular kidney BCRs from control

(PBS) animals, *$P$ = 0.0312. **h, k** Representative BCR network plots capturing *Igh* immune repertoire of extra- and intravascular kidney B cells in control (PBS1) (**h**) and infected (UPEC1) (**k**) mouse. Each vertex represents a unique *Igh* sequence. Edges are generated between vertices that differ by a single nucleotide (non-indel). Clusters/vertices sharing the same CDR3 sequence and present both in extra- and intravascular repertoire are coloured. **i** Representative lineage tree of one *Igh* clone (No. 8) shared between extra- and intravascular kidney compartment (based on identical nucleotide CDR3 sequence) found in control mouse 1. The tree is rooted in the closest germ-line $V_H$ gene allele in the IMGT database (black dot). Each dot represents a unique *Igh* sequence. **j** Tukey box plots comparing total mutation frequency (including silent and replacement mutations) in $V_H$-segments (down-sampled to *n* = 1000 per group/isotype) between extra- and intravascular kidney B cells from control (PBS) mice. *Ighm** represents only sequences with no *Ighd* counterpart (sharing CDR3 nucleotide sequence). **$P$ < 0.01, ***$P$ < 0.001, ****$P$ < 0.0001 and ns $P$ > 0.05. $P$ values were calculated with two-tailed Wilcoxon matched-pairs signed ranked test (**b, c, e, g, j**) and two-tailed Mann-Whitney U test (**e** – comparison between extravascular compartments of PBS and UPEC groups). Source data are provided with this paper.

macrophages and other tissue-resident innate immune cells[65,66], but will require further study to delineate.

In the renal tract, in vivo bacterial challenge with UPEC yielded similar results to that observed in lung infection[46], with μMT⁻ mice demonstrating reduced susceptibility and p110δ^E1020K-B mice increased susceptibility to pyelonephritis and cystitis. We delineate the mechanism, showing that tissue-resident B cells influence macrophage function, at least in part, via IL10. One caveat is that the two genetically modified mouse models we used (uMT⁻ and p110δ^E1020K-B mice) differ from controls not only in their number of IL10-producing tissue-resident B cells but also, for example, in their B-cell signaling, selection and antibody repertoire, which may modulate infection susceptibility[46,51]. However, we found pro-inflammatory transcriptomic and functional changes in tissue macrophage, as well as increased bacterial clearance, in B-IL10-KO mice, mirroring our observations from μMT⁻ animals, confirming the importance of B-cell derived IL10 in shaping macrophage polarization. Our finding that resident B-1 cells were an important cellular source of tissue IL10 both in WT and p110δ^E1020K-B and that the abundance of these cells significantly correlated with the proportion of anti-inflammatory (CD206⁺) macrophages within each organ is consistent with the conclusion that this B cell-mediated macrophage polarization takes place, at least in part, locally. IL10⁺ B-1 cells have been reported in normal skin and liver previously, but their role in organ immunity in homeostasis was not explored[67,68]. Moreover, the conclusions that can be drawn from these studies are substantially limited by their methodology, as they did not use IV labelling premortem to delineate true extravascular tissue B cells[67,68].

In summary, we identified bona-fide tissue-resident B-1 cells across several major non-lymphoid organs in homeostasis using IV labelling and parabiosis, examining their phenotype and function, and demonstrating that they exert a profound effect on macrophage polarization, at least in part, via IL10, and that this ultimately modulates organ susceptibility to bacterial infection (Fig. 8).

## Methods

### Study design

The objective of this study was to examine phenotype and function of B cells residing in mouse and human NLOs in homeostasis. We used IV labelling to identify extravascular B cells in single-cell suspensions from matched murine NLOs (urinary bladder, kidney, liver, lung, peritoneal lavage) with blood and spleen by flow cytometry and BCR sequencing. Parabiosis was used to assess the tissue-residency status of these cells, and immunohistochemistry to delineate their intra-organ distribution and relationship to vasculature and MNPs. We profiled NLOs from multiple mouse models to explore how different genetic background microbiome, age and BCR specificity influence the

tissue-resident B cells compartment. To translate our findings to human, we phenotyped B cells from paired human kidney and spleen samples. To explore the functional impact of B cells on tissue immunity, we compared differences in early response to bacterial challenge of the urinary tract in mice with altered tissue-resident B cell compartment (μMT⁻ and p110δ^E1020K-B) and with B-cell specific IL10 deficiency (*Cd19creIl10^fl/fl*). Finally, we assessed the effect of tissue-resident B cells on kidney MNPs using bulk and sc-RNA-seq, flow-cytometry and phagocytosis assays to assess MNP phenotype, transcriptome and function. In all cases, sample sizes and numbers of replicates for each experiment are indicated in the figure legends. BCR/RNA-seq experiments were performed only once because of their high cost.

### Mouse strains

Wild-type C57BL/6J (stock No. 000664), CD45.1 (stock no. 002014), Balb/c (stock No. 000651), DBA/2J (stock No. 000671), μMT⁻ (stock No. 0002288) and *Mb1^cre* (stock no. 020505) mice were obtained from Jackson Laboratories/Charles River Laboratories (Margate, UK). All animals used were on a C57BL/6 background with the exception of the experiment comparing differences between common wild-type laboratory mouse strains (Balb/c, DBA/2J). P110δ^E1020K-B fl/fl mice were kindly provided by Klaus Okkenhaug (University of Cambridge, UK)[46]. Transgenic mice expressing Venus EYFP under control of the CD11c promoter were a gift from Michel Nussenzweig (Rockefeller University, New York, USA)[69]. Frozen kidneys from *Cd19cre^+/-Il10^fl/fl* mice and their *Cd19^WTIl10^fl/fl* controls were kindly provided by Rudolf Manz (University of Luebeck, Germany). *Cd19cre^+/-Il10^fl/fl* x *Cd19^WTIl10^fl/fl* breeder pair was a gift from Kathryn Else (University of Manchester, UK) and Axel Roers (University of Heidelberg, Germany)[70].

Unless further specified, 8- to 12-week old age- and sex-matched mice were used and co-housed throughout the duration of experiments. In-vivo UTI experiments requiring double urinary bladder catheterization were performed in female mice only because of technical difficulty with male urethral catherization. Mice for non-UTI experiments were not selected for gender. If experiments were not performed on littermates, mice were co-housed for at least three weeks beforehand. The resource equation was used to determine sample size for most experiments. Randomization was genetic and, where possible, investigators were blinded to the genetic status.

All procedures were ethically approved by the University of Cambridge Animal Welfare and Ethical Review Body and complied with the Animals (Scientific Procedures) Act 1986 Amendment Regulations 2012, under the authority of a UK Home Office Licence. Mice were maintained in specific-pathogen-free (SPF) conditions (unless kept as germ-free), on standard diet, at 20–23 °C, with 40–60% humidity, 12-h light:12-h dark cycle at a Home Office-approved facilities in the UK.

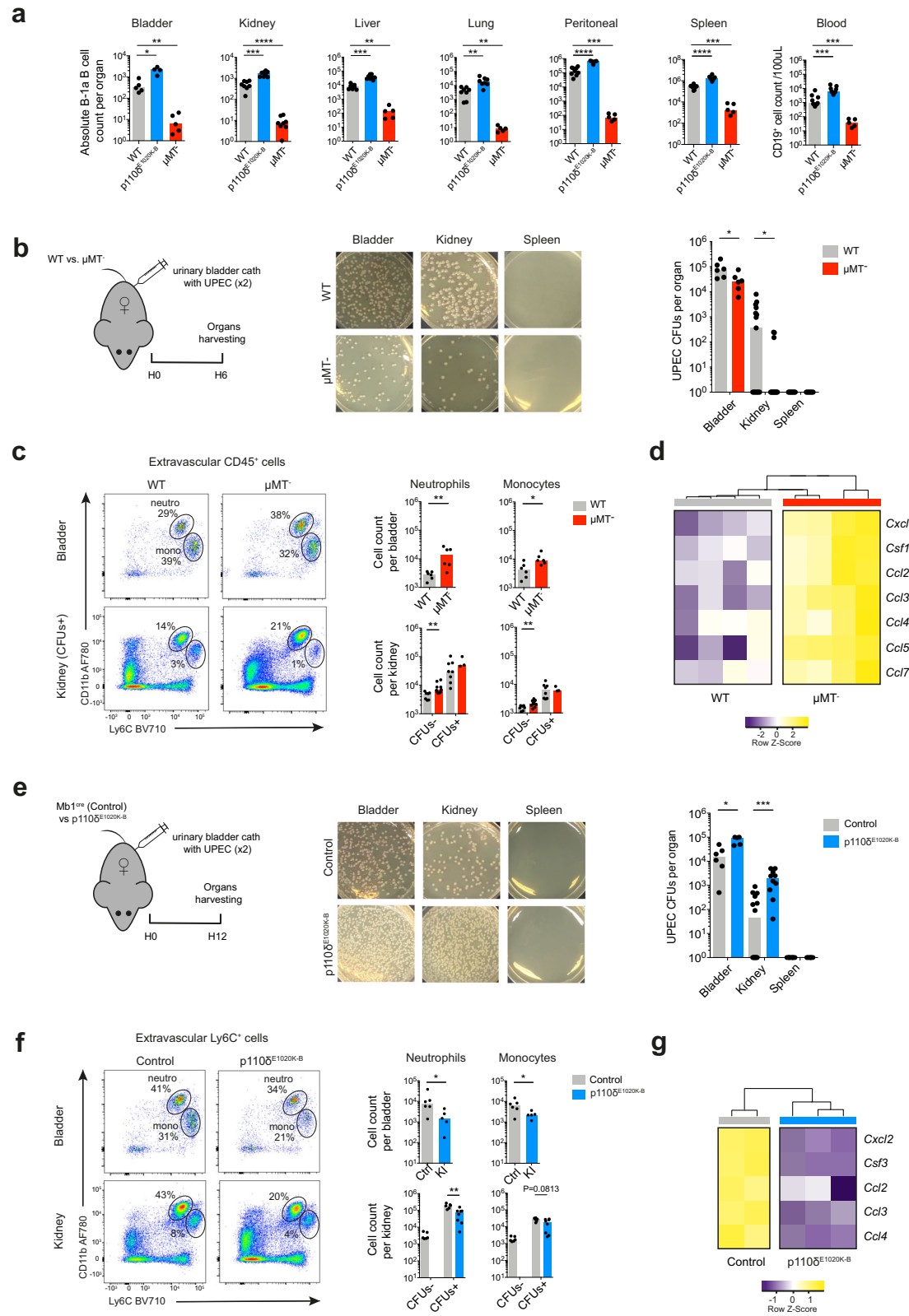

Pet store mice were purchased from various Twin Cities (Minnesota, USA) area pet stores. Female pet store mice were introduced into the cages of 6-10 week old C57BL/6 female mice (National Cancer Institute, USA). Co-housing occurred within a BSL-3 facility, while age-matched C57BL/6 female mice maintained in SPF BSL-1 facilities served as controls as described[43]. To achieve microbial transfer, mice were co-housed with pet store mice for at least 60 days. Co-housing and parabiosis studies were done at the University of Minnesota (USA), in accordance with the guidelines of the Institutional Animal Care and Use Committee (IACUC) at the University of Minnesota (USA).

**Human samples**

Kidneys (with matched splenic tissue for cross-match purposes) donated for transplantation, but unsuitable for implantation due to

**Fig. 5 | Tissue-resident B-1 cells contribute to bacterial defence in the renal tract. a** Flow cytometry quantification of absolute B-1a cells counts found in extravascular organ compartments and blood of WT C57BL/6, p110δ[E1020K-B] and μMT- mice. Bladder (WT $n = 5$, p110δ[E1020K-B] $n = 4$, μMT- $n = 5$): $^*P = 0.0159$, $^{**}P = 0.0079$; kidney (WT $n = 8$, p110δ[E1020K-B] $n = 8$, μMT- $n = 9$): $^{***}P = 0.0002$, $^{****}P < 0.0001$; liver (WT $n = 8$, p110δ[E1020K-B] $n = 8$, μMT- $n = 5$): $^{**}P = 0.0016$, $^{***}P = 0.0002$; lung (WT $n = 8$, p110δ[E1020K-B] $n = 8$, μMT- $n = 5$): p110δ[E1020K-B]$^*P = 0.0019$, $^{μ}$MT $^{**}P = 0.0016$; peritoneal (WT $n = 9$, p110δ[E1020K-B] $n = 8$, μMT- $n = 5$): $^{***}P = 0.0010$, $^{****}P < 0.0001$; spleen (WT $n = 10$, p110δ[E1020K-B] $n = 8$, μMT- $n = 5$): $^{***}P = 0.0007$, $^{****}P < 0.0001$; blood (WT $n = 10$, p110δ[E1020K-B] $n = 9$, μMT- $n = 5$): p110δ[E1020K-B] $^{***}P = 0.0004$, μMT $^{***}P = 0.0007$. Data pooled from two independent experiments. **b, e** Schematic experimental setup (left) for short UTI experiments: μMT- female mice and their matched WT controls (**b**), or p110δ[E1020K-B] female mice and their matched (*Mb1*[cre]) controls (**e**) had urinary bladders inoculated twice (within 45 mins) with UPEC. Their organs were harvested after 6 or 12 hours, respectively. Representative LB agar plates photographs (middle) and UPEC colony forming units (CFU) counts (right) recovered from urinary bladder, kidney and spleen single cell suspensions are shown. A pseudo-count of 1 was added to all CFUs counts. Top (**b**): Bladder ($n = 6$) $^*P = 0.0316$; kidney (WT $n = 15$, μMT- $n = 13$) $^*P = 0.0260$; spleen (WT $n = 8$, μMT- $n = 7$). Bottom (**e**): Bladder (Ctrl $n = 6$, KI $n = 5$) $^*P = 0.0108$; kidney (Ctrl $n = 14$, KI $n = 10$) $^{***}P = 0.0002$; spleen (Ctrl $n = 7$, KI $n = 5$). **c, f** Flow cytometry quantification of absolute extravascular neutrophils (Ly6C[int]CD11b[hi]) and monocytes (Ly6C[hi]CD11b[int]) counts in urinary bladders and kidneys from μMT- mice and their WT controls (**c**), or p110δ[E1020K-B] mice and their controls (**f**) based on experiments described in **b** and **e**, respectively. Top (**c**): Bladder $^{**}P = 0.0087$, $^*P = 0.0260$; kidney neutrophils $^{**}P = 0.0020$, kidney monocytes $^{**}P = 0.0068$. Bottom (**f**): Bladder neutrophils $^*P = 0.0303$, monocytes $^*P = 0.0455$; kidney $^{**}P = 0.0027$. **d, g** Heatmap with hierarchical clustering showing selected significant (*P*adj <0.05) differentially expressed neutrophil- and monocyte-recruiting chemokines genes in kidneys from μMT- mice and their WT controls (**d**), or p110δ[E1020K-B] mice and their controls (**g**) based on experiments described in **b** and **e**, respectively. Data generated from RNA sequencing of whole kidneys positive for UPEC CFUs. UTI experiment data representative of three independent experiments. P values were calculated with two-tailed Mann-Whitney U test and DESeq2 based Wald test with adjustment for multiple testing (**d**, **g**), box plots show medians. Source data are provided with this paper.

damage to the arterial patch or suspicion of donor malignancy (Supplementary Fig. 1b) were used. Ethical approval was granted by the East of England Cambridge Central Research Ethics Committee (REC12/EE/0446) and the study was also approved by NHS Blood and Transplant (NHSBT). Informed consent was obtained from all study participants or their relatives as per study protocol. Kidneys had a cold ischemic time of less than 24 hrs prior to processing. All analyses of human material were performed in the UK. Demographic donor data was retrieved from the NHSBT Electronic Offering System (EOS) files (Supplementary Fig. 1b). Donors for this study were not selected for their gender or sex.

### Bacterial strains
Wild-type UPEC (UTI89, a gift from S. Hultgren, Washington, USA), for in-vivo UTI was prepared following the protocol by Hung et al.[71]. Briefly, frozen bacteria were streaked on a LB agar plate and incubated overnight at 37 °C. On the second day, a single bacterial colony was inoculated into 10uL of LB and cultured statically for 24 hrs at 37 °C. On the third day, 25uL of the bacterial suspension was passaged into fresh 25 mL of LB and cultured statically for 24 hrs at 37 °C again. On the following day, the entire bacterial culture was spun and re-suspended and diluted in sterile PBS to achieve OD600 1.0.

Fluorescent UPEC (vsfGFP-9), a chromosome-based GFP derivate of UTI89, was a gift from S. Chen (National University of Singapore, Singapore)[72]. Its preparation was identical to wild-type UPEC described above with the exception of using low-salt LB (5 g/L NaCl, 5 g/L yeast extract, 10 g/L Tryptone).

### IV labelling and mouse tissue homogenization
To label circulating leukocytes, mice were injected 1.5 μg of anti-CD45-A488 or BV650 antibody (clone: 30-F11, Biolegend, UK) diluted in 200 μL of sterile PBS intravenously three minutes before their euthanasia. Subsequently, a blood sample was obtained by cardiac puncture and transferred into EDTA test tube. Then, peritoneal lavage was performed with 5 mL of cold PBS (with 2% foetal bovine serum, FBS). Upon opening the pleural and peritoneal cavity, whole lungs, liver, spleen, kidneys and urinary bladder were harvested and transferred into ice-cold sterile digest mix. Mouse organs were not perfused.

Digest mix contained RPMI with 0.02 mg/mL (0.03 mg/mL for bladders and lungs, respectively) Liberase TM (Roche), 0.05 mg/mL DNase I (Roche) and 10 mM HEPES (ThermoFisher, UK). Organs were minced with scissors and incubated at 37 °C (rocker incubator) for 30 minutes. After digestion, tissue was homogenized using 70μm Falcon™ Cell strainer (Fisher Scientific, Loughborough, UK) and follow-through suspension quickly washed with up to 10 mLs of cold PBS with 0.5% (w/v) bovine serum albumin (BSA) (Sigma). For splenic, liver, lung, kidney and blood cell suspensions, red cell lysis was performed using distilled H2O containing 0.83% (w/v) NH4Cl, 0.1% (w/v) NaHCO3, 100 μM EDTA. Cell suspensions were then transferred into FACS tubes and 123count eBeads™ (ThermoFisher, UK) counting beads were added to each sample. All samples were spun before staining at 350 x *g* for 5 mins (4 °C) and re-suspended in 100 μL PBS.

### Flow cytometry and sorting
Cell suspensions were pre-incubated with 5 μL human Fc Blocking Reagent (Miltenyi Biotech, Bisley, UK) or normal rat serum (in dilution 1:100) for 10 minutes at 4 °C. Aqua Live/Dead™ 405 nm cell stain (Invitrogen, Paisley, UK) or Zombie UV Aqua Fixable Viability Stain (Biolegend, London, UK) was added in dilution 1:200 in PBS and incubated for 5 minutes, followed by the surface antibody cocktail for 30 mins in the dark at 4 °C. All rat (anti-mouse) and mouse (anti-human) antibodies were used in dilution 1:100 and 1:25, respectively. All antibodies used in this study are listed in Supplementary Table 1. Intranuclear Ki-67 staining was performed using eBioscience™ Foxp3/Transcription Factor Staining Buffer Set (ThermoFisher Scientific, Waltham, USA).

Following three washes, samples were measured with BD LSRFortessa using BD FACSDiva 6.2 software (Becton Dickinson, Basel, Switzerland) and data subsequently analysed with FlowJo 10.4 software (TreeStar, USA). Each FACS tube was run until its exhaustion. Cell sorting of murine kidney and bladder MNPs and B cells was done on unfixed cells using Aria-Fusion III (Becton Dickinson, Basel, Switzerland) or iCyt Synergy (Sony Biotechnology Inc.).

### Immunofluorescence staining for confocal microscopy
Murine tissue samples were fixed with 1% (*w/v*) paraformaldehyde (Electron Microscopy Services) in PBS for 16 h, washed with PBS, and equilibrated in 30% (*w/v*) sucrose for a further 16 h. Tissues were then frozen at −80 °C in Optimal Cutting Temperature (OCT) embedding medium (Thermo Fisher Scientific). 20um cryostat sections air-dried for 30 min and then permeabilised and blocked with blocking buffer (0.1 M TRIS), containing 0.1% Triton X-100 (Sigma), 1% (*w/v*) normal rat serum, 1% (*w/v*) BSA (R&D Systems, Abingdon, UK) for 1 hour at RT. Sections were stained overnight at 4 °C with a combination of the following antibodies in blocking buffer: CD19 (6D5; Biolegend, dilution: 1:25, RRID: AB_11218994, Cat. No. 115543), CD31 (MEC13.3; Biolegend, dilution: 1:50, RRID: AB_2563319, Cat. No. 100237), CD5 (53-7.3; Biolegend, dilution: 1:25, RRID: AB_2563930, Cat. No. 100627).

Slides were mounted using Vectashield Mounting Medium without or with DAPI (Vector Laboratories Inc., USA). Images were acquired using TCS SP8 (Leica, Milton Keynes, UK) confocal microscope with ZEN 3.7 software (Zeiss, Cambourne, UK). Raw imaging data were processed and quantified using Imaris v9.3 (Bitplane, Zurich, Switzerland).

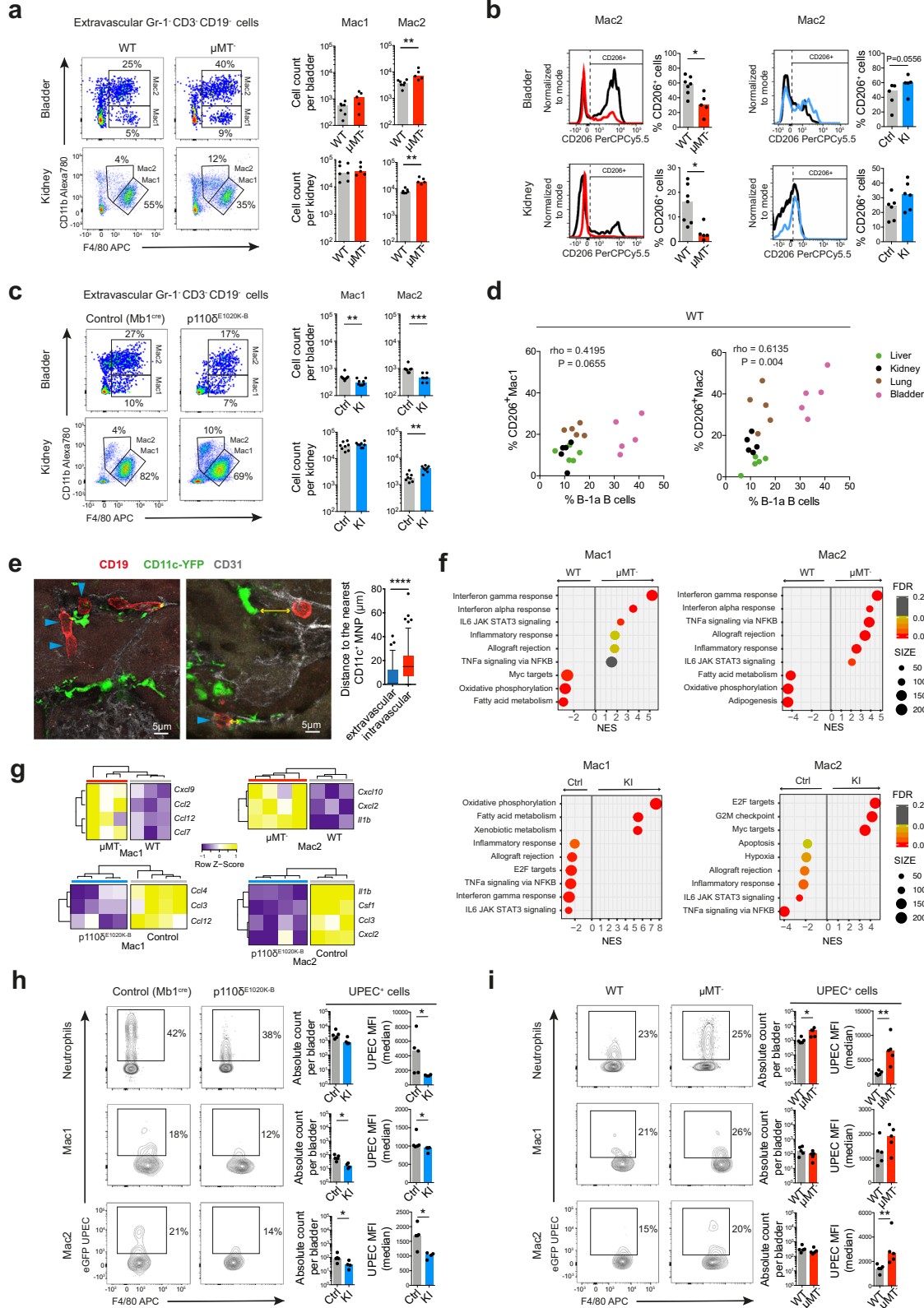

## Intracellular IL10 staining

Single cell suspensions obtained from homogenized murine kidneys, livers and spleen were subjected to 44% (v/v) Percoll gradient centrifugation (GE Healthcare Life Sciences, Little Chalfont, UK) and washed twice with ice-cold PBS before stimulation. Cell suspensions from urinary bladders and lungs were not purified by Percoll gradient because of limited leukocyte yield.

$5 \times 10^5$ cells were re-suspended in 500 µL of complete RPMI (Sigma-Aldrich, Gillingham, UK) with 50 ng/mL Phorbol 12-myristate 12-acetate (PMA) (Sigma-Aldrich, Gillingham, UK), 1 ug/mL ionomycin (Sigma-Aldrich, Gillingham, UK) and BD GolgiStop™ (BD Biosciences). The suspensions were vortexed quickly and incubated for 4 hours at 37 °C with 5% $CO_2$. At the end of the stimulation, cells were washed with ice-cold PBS once and stained for surface antigens as described above.

**Fig. 6 | Tissue-resident B cells orchestrate macrophage polarisation in the renal tract. a, c** Representative flow cytometry plots with gating Ly6C⁻F4/80ʰⁱCD11bˡᵒʷ (Mac1) and Ly6C⁻F4/80ˡᵒʷCD11bʰⁱ (Mac2) MNPs in the extravascular compartment of urinary bladders (top) and kidneys (bottom) at steady state from μMT⁻ mice (*n* = 5) and their matched WT controls (*n* = 7) (**a**), or p110δ^E1020K-B^ mice (*n* = 8) and their *Mb1ᶜʳᵉ* controls (*n* = 8) (**c**), respectively. Bar charts show absolute counts of these MNPs subsets per organ. Data representative of three independent experiments. μMT⁻ mice: Bladder **P = 0.0051; kidney **P = 0.0051; p110δ^E1020K-B^ mice: Bladder **P = 0.0044, ***P = 0.0002; kidney **P = 0.0011. **b** Representative histograms showing CD206 MFI of Mac2 analyzed in **a** and **c**. Bar charts quantify percentage of CD206⁺ cells within each subset. μMT⁻ (*n* = 5)/WT (*n* = 7) mice *P = 0.0177; p110δ^E1020K-B^/control mice: bladders (*n* = 5 per group), kidneys (*n* = 6 per group), *P = 0.0044, ***P = 0.0002. **d** Correlation between the percentage of B-1a B cells (within extravascular B cells) and the percentage of CD206⁺ cells within Mac1 (left) or Mac2 (right) subsets, respectively, found in WT C57BL/6 mouse liver, kidney, lung and urinary bladder (*n* = 5 per organ). **e** Confocal microscopy of B cells (CD19, red), MNPs (CD11c, green) and endothelium (CD31, grey) in renal cortex from 10-week old CD11c-YFP reporter mice at steady state. Tukey box plot compares the median distance (yellow arrow) of extra- versus intravascular B cells to the nearest CD11c⁺ MNP, *n* = 100 cells per group, ***P < 0.0001. **f** Hallmark GSEA on pre-ranked genes differentially expressed in sorted extravascular Mac1 (F4/80ʰⁱCD11bˡᵒʷ, left plots) and Mac2 (F4/80ˡᵒʷCD11bʰⁱ, right plots) subsets, respectively, from kidney single-cell suspensions stimulated with LPS for 2 hrs and obtained from either μMT⁻ (*n* = 4, upper plots), or p110δ^E1020K-B^ (*n* = 4, lower plots) mice, respectively, and their matched controls (WT *n* = 3, *Mb1ᶜʳᵉ* controls *n* = 3). Top six positively and top three negatively enriched Hallmark gene sets shown. NES−normalized enrichment score, FDR−false discovery rate. **g** Heatmap showing significant (*P*adj < 0.05) neutrophil- and monocyte-recruiting chemokines genes differentially expressed in the same μMT⁻ (upper plots), or p110δ^E1020K-B^ (lower plots) mouse kidney macrophages datasets as described in **f**. **h, i** Flow cytometry evaluation of in-vivo eGFP UPEC phagocytosis by extravascular neutrophils, Mac1 and Mac2 in p110δ^E1020K-B^ (KI, *n* = 4) (**h**) and μMT⁻ (*n* = 5) (**i**) urinary bladders, respectively, and their controls (*Mb1ᶜʳᵉ* controls *n* = 5, WT *n* = 5), 6 hours after inoculation. Bar charts show absolute counts of eGFP UPEC⁺ cells per urinary bladder (left) and their median eGFP MFI (right). Data representative of two independent experiments. Detailed gating strategy shown in Supplementary Fig. 6f, g. μMT⁻ mice: Mac2 *P = 0.0317, others *P = 0.0159; p110δ^E1020K-B^ mice: *P = 0.0317, **P = 0.0079. *P* values were calculated with two-tailed Mann-Whitney U test (**a**–**c**, **e**, **h**, **i**), two-tailed Spearman correlation (**d**) or DESeq2 based Wald test with adjustment for multiple testing (**f, g**). Box plots show medians shown unless stated otherwise. Source data are provided with this paper.

Following intracellular staining for IL-10 was performed using BD Cytofix/Cytoperm™ kit (BD Biosciences) and anti-mouse IL-10 antibody at dilution 1:100 (JES5-16E3; BD Pharmigen), or isotype control (MPC-11; Biolegend).

### 5-ethynyl-2'-deoxyuridine (EdU) proliferation assay
Drinking water with 0.5 mg/mL EdU (Glentham Life Sciences, Corsham, UK) and 1% sucrose (Sigma-Aldrich, Gillingham, UK) was given to experimental animals for up to four weeks. EdU solution was made up freshly, protected from light and exchanged every three days. At the end of the experiment, mouse organs were harvested and homogenized as outlined above. EdU incorporated into DNA of the proliferating cells was labelled using Click-iT™ EdU Alexa Fluor™ 647 Flow Cytometry Assay Kit (ThermoFisher Scientific, Waltham, USA) and detected by flow cytometry.

### Murine B cell depletion
Wild-type C57/BL6 mice were treated with a single dose of 10 mg/kg anti-mouse CD20 depleting antibody (clone 5D2, generous gift from Genetech, Montreal, Canada), or control antibody (rituximab/Truxima™, Healthcare CellTrion) administered intravenously. After seven days, peripheral blood sample was obtained to confirm successful depletion. Mice were euthanised and their blood and organs processed four weeks after administration of depleting antibody.

### LPS stimulation, RNA extraction and reverse transcription
Murine kidney single cell suspensions were stimulated in complete RPMI with 100 ng/mL LPS (Sigma-Aldrich) for 2 hrs at 37 °C and 5% CO₂. After two washes with ice-cold PBS, cells were either stained for sorting (RNA-seq), or re-suspended in 350 μL of RTL Buffer (Qiagen, Manchester, UK) and processed with QIAshredder homogenizer (Qiagen, Manchester, UK). The follow-through was immediately frozen and stored at −80 °C for not more than two weeks before further processing.

RNA was extracted from thawed samples using Ambion RNA PureLink Kit (Life Technologies, Paisley, UK) and RNA yields analyzed by NanoDrop spectrophotometer (Thermo-Scientific, Loughborough, UK). Complementary DNA (cDNA) was prepared by using High Capacity RNA-to-cDNA™ Kit (Life Technologies, Paisley, UK) and BioRad PCR machine (BioRad, Hemel Hempstead, UK).

### Quantitative polymerase chain reaction
RT-PCR was performed using TaqMan™ 2X Fast Master Mix on the Viia 7 PCR machine (Life Technologies, Paisley, UK). The following TaqMan gene expression primers (Life Technologies, Paisley, UK) were used for the reaction Mm01288386_m1 (*Il10*), Mm99999915_g1 (*Gapdh*), Mm00443258_m1 (*Tnfa*). Gene expression relative to *Gapdh* was calculated using $2^{-\Delta Ct}$ as described previously[73].

### Bulk RNA-seq samples preparation
Murine kidney single-cell suspensions stimulated with LPS as described above were flow-sorted at 4 °C for live CD45(IV)⁻CD45⁺CD19⁻CD3⁻ MNP1 (F4/80ʰⁱCD11bˡᵒʷ) and MNP2 (F4/80ˡᵒʷCD11bʰⁱ). Cells were sorted directly into 750 μL RLT plus buffer (QIAGEN), immediately vortexed, snap frozen on dry ice and stored at −80 °C. RNA was extracted from cell lysates using the RNeasy plus micro kit (QIAGEN) as per manufacturer's instructions. Optimal DNA depletion columns (QIAGEN) were used to remove contaminating genomic DNA. Purified RNA was eluted in nuclease free water (Ambion) and stored at −80 °C. Quality and concentration of the purified RNA was assessed using an RNA Pico Chip (Applied Biosystems) using a Bioanalyzer 2000 (Applied Biosystems) as per the manufacturer's instructions. For all RNA-seq experiments, samples had an RNA integrity number greater than 8. For the preparation of libraries, SMARTer stranded total RNA-Seq Mammalian Pico Input kit (Takara) was used as per manufacturer's instructions. To produce the libraries, 1.5-3.55 ng of total RNA was used and libraries were amplified for 14 cycles of PCR. Library size was assessed using 1 μL of undiluted final libraries with a High Sensitivity DNA Chip (Applied Biosystems) using a Bioanalyzer 2000 (Applied Biosystems) as per manufacturer's instructions. Library concentration was quantified by PCR using 1/10000 dilution of the library in nuclease free water (Ambion) with ROX low KAPPA library quantification kit (KAPPA Biosystems). Libraries were pooled at an equimolar concentration with up to 10 libraries per pool.

### Single-cell RNA-seq samples preparation
Frozen unchallenged kidneys (freezing medium: 10% dimethylsufoxide in fetal calf serum (Sigma-Aldrich, Gillingham, UK)) from *Cd19cre⁺/⁻Il10ᶠˡ/ᶠˡ* mice (*n* = 5) and their *Cd19ᵂᵀIl10ᶠˡ/ᶠˡ* controls (*n* = 5) were thawed quickly to 20 °C and homogenized as described earlier. Obtained single cell suspensions were further refined by using 44% Percoll® (GE Healthcare Life Sciences, Little Chalfont, UK) separation gradient and sorted at 4 °C for live single CD45⁺ cells. Sorted samples within each study group were subsequently pooled and their count adjusted to 1 × 10⁶ cells/mL. One lane per study group was loaded according to the standard protocol of Chromium single 3' (V2 chemistry, 10X Genomics, USA) to capture 10,000 cells/channel. Libraries

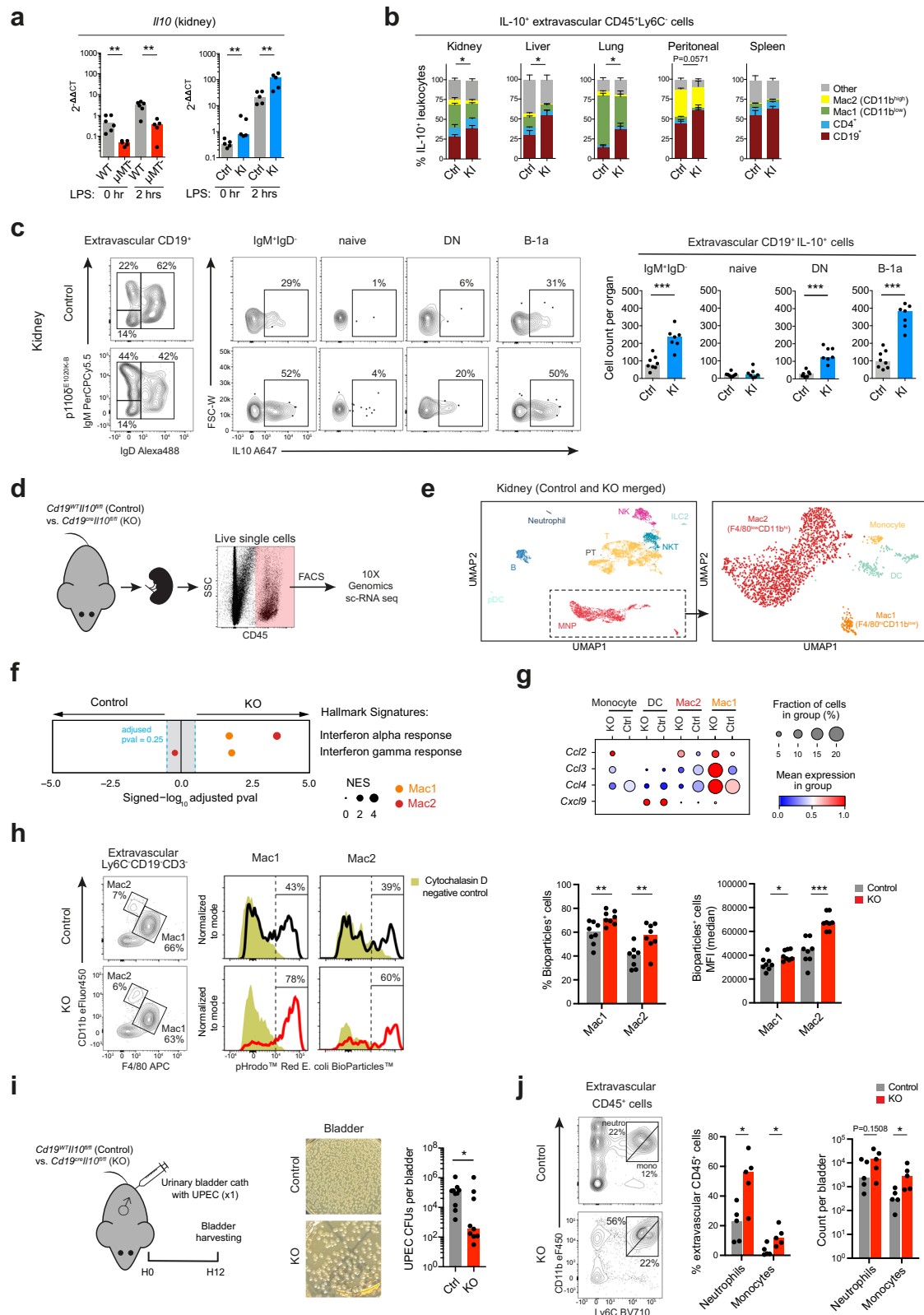

were prepared according to the manufacturer's protocol, followed by Bioanalyzer quality checks.

**BCR-seq samples preparation**
Murine kidney single-cell suspensions from UTI experiment were flow-sorted for live extravascular (CD45IV⁻) or intravascular (CD45IV⁺) B cells (CD45⁺CD3⁻CD19⁺). Sorted B cells RNA was prepared

as described above. Reverse transcription (RT) was performed in two steps: First, 1 uL barcoded reverse primer mix (10 uM per each primer), 1 uL dNTP (10 mM) and 14uL RNA template (up to 500 ng) were mixed and heated to 65 °C for 5 mins followed by an immediate transfer to ice for at least one minute. Second, mixture 2 including 4 uL 5X First-Strand Buffer, 1 uL DTT (0.1M), 1 uL RNaseOUT™ (Thermo Scientific) and 1 uL SuperScript®III (Life Technologies) was

**Fig. 7 | Tissue-resident B cells in non-lymphoid organs are an important source of IL10 regulating local tissue immunity. a** *Il10* mRNA expression by quantitative PCR in whole kidney single-cell suspensions from μMT⁻ (left) or p110δ^E1020K-B (right) mice (*n* = 5 per group), respectively, and their matched controls (WT *n* = 6, *Mb1^cre* control *n* = 5) at baseline and after 2-hr stimulation with LPS. μMT⁻ kidneys: 0-hr **P = 0.0043, 2-hr **P = 0.0087; p110δ^E1020K-B kidneys **P = 0.0079. **b** Proportion of five cell subsets (CD19⁺, CD4⁺, Mac1, Mac2 and other) within IL10-producing extravascular CD45⁺Ly6C⁻ leukocytes from *Mb1^cre* (control, Ctrl, *n* = 6) and p110δ^E1020K (KI, *n* = 6) mouse organs. The statistical testing compares the difference in CD19⁺ cell proportions. Kidney *P = 0.0325; liver & lung *P = 0.0206; peritoneal *P = 0.0571. Data pooled from two independent experiments. **c** Flow cytometry quantification of IL10 expression in extravascular kidney B cell subsets (IgM⁺IgD⁻, naïve, DN, B-1a) from *Mb1^cre* (control, Ctrl, *n* = 8) and p110δ^E1020K (KI, *n* = 7) mice. Bar charts show absolute cell counts IL10⁺ B cells within each subset. Data representative of three independent experiments. IgM⁺IgD⁻ ***P = 0.0006; DN ***P = 0.0006; B-1a ***P = 0.0003. **d** Schematic for the scRNA-seq experiment setup: Live CD45⁺ cells were FACS-sorted from single-cell suspensions of unchallenged kidneys from *Cd19^cre Il10^fl/fl* (KO) mice and their (*Cd19^cre*) controls (Ctrl) (*n* = 5 per group), and subjected to scRNA sequencing (10X). **e** UMAP plots show all sorted cells (left, *n* = 6538) and MNPs-only (right, *n* = 1894) coloured according to major cell type annotations. **f** GSEA for Hallmark 'interferon alpha' and 'interferon gamma' response based on pre-ranked genes differentially expressed in Mac1 (F4/80^hiCD11b^low) or Mac2 (F4/80^lowCD11b^hi), respectively, obtained from control or KO

kidneys described in **d**. NES – normalized enrichment score. **g** Mean expression dot plot of monocyte-recruiting chemokine genes in MNPs showed in **e**, split by experimental group. **h** In-vitro phagocytosis of pHrodo™ Red E. coli BioParticles™ by both macrophage subsets from control (*Cd19^+Il10^fl/fl*) and KO (*Cd19^cre Il10^fl/fl*) kidneys (*n* = 8 per group). Representative flow cytometry plots show gating of extravascular Ly6C⁻F4/80^hiCD11b^low (Mac1) and Ly6C⁻F4/80^lowCD11b^hi (Mac2) MNPs (left). Representative histograms (normalized to mode) display E. coli bioparticles MFI and gating of bioparticles⁺ Macs (right). Dark yellow histograms represent cytochalasin D treated negative controls. Bar charts show percentage of bioparticles⁺ cells (left) and their median MFI (right). %Bioparticles⁺ cells: Mac1 **P = 0.0043, Mac2 **P = 0.0070; MFI: Mac1 *P = 0.0148, Mac2 *P = 0.0002. Data combined from two independent experiments. **i** Schematic for the UTI experiment setup (left): Urinary bladders of *Cd19cre^+/- Il10^fl/fl* (KO, *n* = 9) male mice and their matched controls (*n* = 10) were inoculated once with UPEC and harvested after 12 hours. Representative LB agar plates photographs (middle) and UPEC CFU counts (right) recovered from urinary bladders shown. Data combined from two independent experiments, *P = 0.0220. **j** Representative flow cytometry plots and quantification of extravascular neutrophils (Ly6C^intCD11b^hi) and monocytes (Ly6C^hiCD11b^int) in urinary bladders treated as described in **i**. Data representative of three independent experiments, *P = 0.0317. *P* values were calculated with one-tailed (**b**) and two-tailed (**a**, **c**, **h**, **i**, **j**) Mann-Whitney U test. Medians shown unless stated otherwise. Source data are provided with this paper.

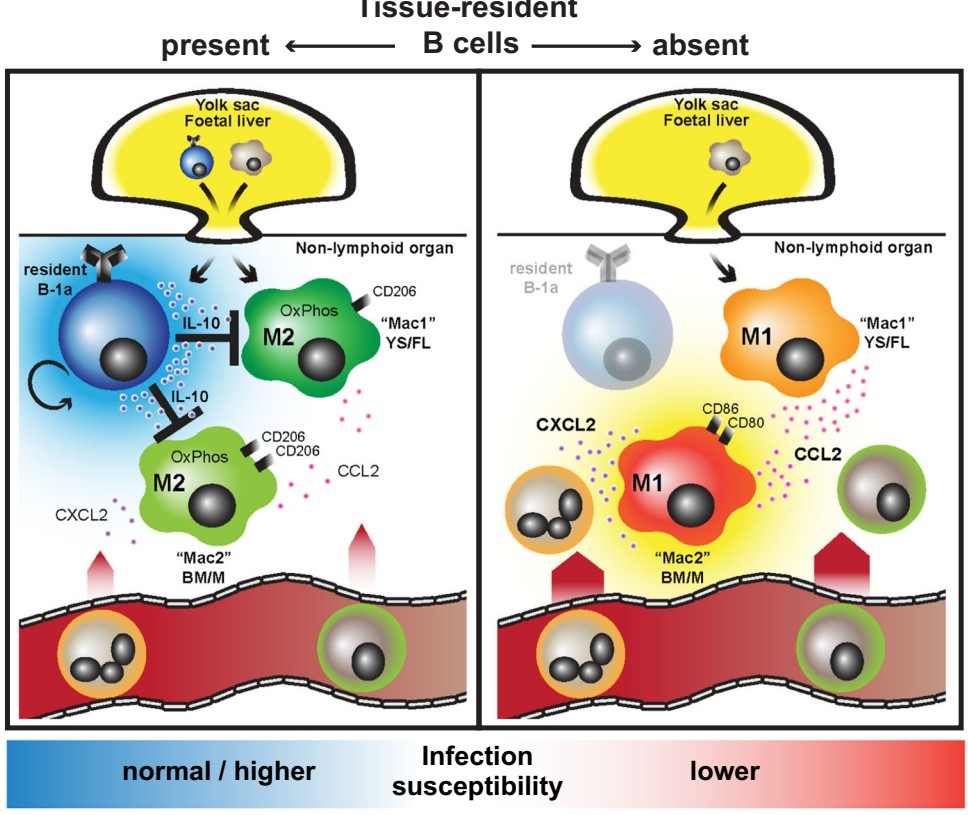

**Fig. 8 | Graphical abstract - Tissue-resident B cells orchestrate macrophage polarisation and function.** Homeostatic seeding of B-1 cells is not limited to body cavities and the spleen, but extends to major non-lymphoid organs, in an analogous way to macrophages. These two cell types reside side by side in tissue niches, enabling B cells to shape macrophage polarisation, at least in part, via IL10 secretion and to set their 'inflammatory set-point', with important consequences for tissue immunity and defence.

added and incubated at 50 °C for 60 min followed by 15 min inactivation at 70 °C. cDNA was cleaned-up with Agencourt AMPure XP beads and PCR amplified with V-gene multiplex primer mix (10 μM each forward primer) and 3′ universal reverse primer (10 μM) using KAPA protocol (KAPA Biosystems) and the thermal cycling conditions: 1 cycle (95 °C−5 min); 5 cycles (98 °C−5 s, 72 °C−2 min); 5 cycles (98 °C−5 s, 65 °C−10 s, 72 °C−2 min); 30 cycles (98 °C−20 s,

60 °C−1 min, 72 °C−2 min) and 1 step (72 °C−7 min). Primers are provided in Supplementary Table 2.

### In-vivo murine urinary tract infection (UTI) and eGFP UPEC phagocytosis
Urinary tract infection was induced in experimental mice as previously described[71,74]. Briefly, under isoflurane anesthesia (Baxter, Newbury,

UK), the perineum was cleaned with ethanol, the urethra catheterized using $0.28 \times 0.60$ mm polyethylene tubing (Instech Laboratories, PA, USA) lubricated in sterile Instillagel (CliniMed Ltd, UK). $4 \times 10^7$ CFUs in 100uL of prepared WT or eGFP UPEC were inoculated into a bladder during one session. Repeated inoculation was performed within 45 min (or after in 3 hrs if organs harvested in more than 24 hrs) to increase the likelihood of ascending pyelonephritis. For studying UPEC phagocytosis in urinary bladders, a single inoculation with $2 \times 10^7$ in 50uL of eGFP UPEC was used to prevent reflux of the inoculum into kidneys or its leak per urethram.

Mice were euthanised at a specified time point (6 hr, 12 hrs or 6 weeks) depending on the experimental protocol (see figure legends). Mouse organs were harvested and processed as described above. CFUs were quantified using the aCOLyte-3 colony counter (Synbiosis, Cambridge, UK).

## In-vitro phagocytosis assay

Kidneys from *Cd19cre*$^{+/}$*Il10*$^{fl/fl}$ 8-12-week old male mice and their matched controls were harvested and homogenized as described above. Obtained single cell suspensions were further refined by using 44% Percoll® (GE Healthcare Life Sciences, Little Chalfont, UK) and pre-incubated at 37 °C and 5% $CO_2$ for 15 mins. For negative controls, cytochalasin D (Cayman Chemical, Ann Arbor, Michigan, USA) was added to the cell suspensions at the beginning of the pre-incubation (final concentration of 10 μM). Then, the phagocytosis assay was performed using pHrodo™ Red E. coli BioParticles™ (ThermoFisher Scientific, Waltham, USA) as per manufacturer instructions with the incubation time set for 60 mins. At the end of the incubation, samples were washed twice with ice-cold PBS and processed further for flow-cytometry staining and analysis (at 4 °C) as described above.

## Parabiosis

Parabiosis surgeries between age- and sex-matched animals were performed at University of Minnesota (USA) as described previously[75]. Briefly, mice were anesthetized with Avertin (250 mg/kg) by intraperitoneal injection. The corresponding lateral aspects of mice were shaved and skin was disinfected with betadine solution. Skin incisions were made from the shoulder to the hip and the subcutaneous fascia was dissected to create about 0.5 cm of free skin. The dorsal and ventral aspects of the incised skin of corresponding mice were joined by surgical staples. The duration of parabiosis was 28–35 days before euthanasia. Surgeries were conducted in accordance with the guidelines of the IACUC at the University of Minnesota (USA).

## Human tissue processing

Human tissue was weighed, minced with a razor blade and digested with 30 μg/mL Liberase TM, 50 μg/mL DNase I (all from Roche, Burgess Hill, UK) in RPMI for 30 minutes at 37 °C (rocker incubator). After digestion, tissue was transferred into gentleMACS™ C-tubes and dissociated using programme 'Spleen-0' and 'Lung-02' on gentleMACS™ Dissociator (Miltenyi Biotech, Bisley, UK). Homogenized tissue was filtered through 100-μm and 40-μm Falcon™ cell strainers (Fisher Scientific, Loughborough, UK). After one wash in running buffer (PBS, 2 mM EDTA, 0.5% (w/v) BSA), the cell suspension was further refined by using 44% Percoll® (GE Healthcare Life Sciences, Little Chalfont, UK) separation gradient and Dead cell removal kit (Miltenyi Biotech, Bisley, UK).

Human splenic tissue was weighed and then homogenized using 70-μm Falcon™ cell strainer (Fisher Scientific, Loughborough, UK) in a running buffer. Repeated red cell lysis was then performed as described in murine experiments. Human peripheral mononuclear cells (PBMCs) were isolated from 9 mL of full blood (with EDTA anticoagulant) using Histopaque®-1077 separation gradient (Sigma-Aldrich, Gillingham, UK). After cell counting, splenic suspension or PBMCs were directly stained for FACS/cytometry without any further purification steps.

## Quantification and statistical analysis

Statistical analyses were performed using GraphPad Prism 9 software (La Jolla, USA). Error bars and statistical tests used in each experiment and the adjustments for multiple comparisons are specified in figures description. Biological replicates are displayed as individual data points and their number ($n$) specified in figure legends. $^*P < 0.05$, $^{**}P < 0.01$, $^{***}P < 0.001$, $^{****}P < 0.0001$, ns if $P > 0.05$. Medians are shown in bar charts unless stated otherwise.

## Confocal image analysis

Confocal tile scan images were imported into Imaris v.9.3 (Bitplane). CD19+ cells were manually labelled either as intra-, or extravascular based on their 3D position in relation to CD31+ vessels. Their distance to the nearest CD11c$^+$ mononuclear phagocyte (MNP) was measured manually (surface to surface).

For 2D distribution (clustering) analysis, separate masks were created for extravascular B cells (manually, using Spots function) and MNPs (using value-based Surface function for detection for all CD11c$^+$ cells). Individual masks were imported into ImageJ v1.53c[76], converted into 8-bit image and particles / objects distribution analysed using 2D Particle Distribution function from BioVoxxell toolbox (v2.5.3) (with default setting)[77]. Nearest neighbour distance (NND) index was calculated for each image as measured median nearest neighbour distance divided by theoretical random nearest neighbour distance.

## Bulk RNA sequencing and analysis

Sequencing of the libraries was carried out using a Novaseq 4000 (Illumina) on a 2 x 150 bp sequencing run with 1 pool per flow cell lane. Sequencing was carried out at Genewiz (NJ, USA). Pooled libraries were de-multiplexed by Genewiz using Casava v1.8 (Illumina) before transfer of the data to University of Cambridge. Fastq files were trimmed of the first three nucleotides of the R1 strand and contaminating adaptor sequences and poor-quality bases removed (bases with a Phred 33 score of <30) using Trimgalore! (v0.6.4) (Babraham Bioinformatics) and quality of the resulting files was assessed using FastQC (v0.11.9) (Babraham Bioinformatics). Fastq files were aligned to the mm10 genome using HISAT2 (v2.1.0)[78]. All analyses were carried out using R version 3.5.2. Reads were counted and assigned to genes using the featureCounts function from the RSubread package (v1.32.0)[79]. Differential expression analysis was carried out using DESeq2 (v1.22.1)[80] using a linear model with an appropriate design matrix following the default workflow (with betaPrior = FALSE) and subsequent application of lfcShrink (type = apeglm)[81]. Adjusted $P$ value < 0.05 was considered as significant.

Resulting figures were plotted using ggplot2 and heatmap.2 from the gplots package. GSEA was performed for RNA-seq data by first assigning a rank metric to each gene using the following formula: rank = $(1/(P\text{-value} + 1 \times 10^{-300}))*(|LFC|/LFC)$. GSEA was then run using GSEA (v3.0)[82] using the pre-ranked option with the classic setting against Hallmarks gene set from the molecular signature database or a macrophage stimulation gene set. The macrophage stimulation gene set was generated from publicly available sequencing data (GSE47189)[57] based top 100 genes upregulated when compared to baseline for each stimulation condition.

## Single-cell RNA sequencing and analysis

Sequencing was performed on an Illumina Hiseq 4000 at Genewiz (NJ, USA). Single-cell gene expression data from cellranger (v5) output were analysed using standard Seurat-inspired scanpy (v.1.7.7) workflow[83,84]. Doublet detection was performed using scrublet (v0.2.3)[85] with adaptations outlined in[86]. Briefly, after scrublet was performed, the data were iteratively sub-clustered and a median scrublet score for each sub-cluster was computed. Median absolute deviation (MAD) scores were computed from the cluster scrublet scores and a one tailed t-test was performed with Benjamini-Hochberg

(BH) correction[87] applied and cells with significantly outlying cluster scrublet scores (BH pval <0.1) were flagged as potential doublets. The data were then processed using scanpy with standard quality control steps; cells were filtered if number of genes > 6000 or <200. Only droplets with unique molecular identifier (UMI) count of >=1000 were retained for analysis. Mitochondrial content cut-off was determined by fitting a two-component Gaussian mixture model to UMI counts contributed by mitochondrial genes (gene starting with Mt-) and total UMI counts per cell, clustering cells into either high or low mitochondrial content clusters; low mitochondrial content clusters were retained for further analyses. Genes were retained if they are expressed by at least 3 cells. Genes counts for each cell were normalized to contain a total count equal to 10000 counts per cell. This led to a working dataset of 6,538 cells. Highly variable genes were selected based on the following parameters: minimum and maximum mean expression are >=0.0125 and <=3 respectively; minimum dispersion of genes = 0.5. The number of principal components used for neighborhood graph construction and dimensional reduction was set at 40. Clustering was performed using Leiden algorithm[88] with resolution set at 1.0. Uniform Manifold Approximation and Projection (UMAP; v3.10.0)[89] was used for dimensional reduction and visualization, the minimum distance was set at 0.3 and all other parameters as per default settings in scanpy. Cell type identification was guided by manual inspection of top marker genes identified with Wilcoxon rank sum tests.

Gene set enrichment analysis was performed using fgsea package available on Bioconductor and visualized with the GOChord function in the GOplot package (v1.0.2). Briefly, genes were ranked in the descending order by the Wilcoxon statistic value from the pairwise Wilcoxon rank sum tests. The top ten leading-edge genes from the 'interferon alpha response' and 'interferon gamma response' pathways were then subject to visualisation.

Publically available sc-RNA seq dataset of wild-type murine kidneys (GSE175792) was processed through the same quality control steps as described above, leading to 106,531 cells available for downstream analysis.

### BCR-sequencing and analysis
MiSeq libraries were prepared using Illumina protocols and sequenced using 300 bp paired-ended MiSeq (Illumina). Raw MiSeq reads were filtered for base quality (median Phred score > 32) using QUASR 6.X (http://sourceforge.net/projects/quasr/)[90]. MiSeq forward and reverse reads were merged together if they contained identical overlapping regions of >50 bp, or otherwise discarded. Universal barcoded regions were identified in reads and orientated to read from V-primer to constant region primer. The barcoded region within each primer was identified and checked for conserved bases (i.e., the T's in NNNNTNNNNTNNNNT). Primers and constant regions were trimmed from each sequence, and sequences were retained only if there was >80% sequence certainty between all sequences obtained with the same barcode, otherwise discarded. The constant region allele with highest sequence similarity was identified by 10-mer matching to the reference constant region genes from the IMGT database (v3.1.15)[91], and sequences were trimmed to give only the region of the sequence corresponding to the variable (V–D–J) regions. Isotype usage information for each *Igh* was retained throughout the analysis hereafter. Sequences without complete reading frames and non-immunoglobulin sequences were removed and only reads with significant similarity to reference *Ighv* and *Ighj* genes from the IMGT database were retained using BLAST (v2.7.1)[92]. Only sequences with more than one read were retained for further downstream analysis. *Ighv* and *Ighj* genes and mutational status were determined for each BCR using IMGT/V-QUEST (v3.4.15)[93].

Isotype frequencies were calculated for each sample as proportion of unique *Ighv-d-j* regions per isotype in all unique *Ighv-d-j* regions. SHM levels (including silent and non-silent mutations) per unique *Ighv-*

*d-j* region per isotype were calculated for each individual sample using observedMutation function within SHazaM package (v0.2.1)[94]. The network generation algorithm and network properties were calculated as previously described[95]. Briefly, each vertex represents a unique sequence. Edges are generated between vertices that differ by single-nucleotide, non-indel differences and clusters are collections of related, connected vertices. Lineage trees were generated using build-PhylipLineage function within Alakazam package (v0.3.0)[94] after merging sequences from paired intra- and extravascular samples with identical CDR3. $V_H$ segment usage, mean insert size (i.e., mean number of random nucleotides inserted in CDR3 sequence at V-D and D-J site) and heatmap tracking shared clonotypes across samples were generated using VDJtools (v1.2.1) with default settings. Unique clonotypes numbers used to compare BCR diversity between study groups were normalized to 244 reads (i.e., lowest number of reads per sample observed). Statistics were performed in R using two-tailed Wilcoxon matched-pairs tests for significance (non-parametric test of differences between distributions).

### Reporting summary
Further information on research design is available in the Nature Portfolio Reporting Summary linked to this article.

## Data availability
The accession codes for the sequencing data generated in this paper are GSE139851, GSE139852, GSE139854, GSE139855, GSE139848, GSE139849, GSE184794 and GSE244459. Publicly available sc-RNA seq datasets used in this paper are accessible under GSE175792 and GSE47189. Source data are provided with this paper as a Source Data file. Source data are provided with this paper.

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

## Acknowledgements

The Clatworthy Lab is based in the University of Cambridge Molecular Immunity Unit in the MRC Laboratory of Molecular Biology and is grateful for the use of the core facilities. O.S. was supported by the Wellcome PhD Clinical Training Fellowship 205250/Z/16/Z and National Institute of Health Research (NIHR) clinical lectureship. S.C. and K.H. were supported by Wellcome Trust Research grant 206194. J.R.F. and M.R.C. were funded by the NIHR Cambridge Biomedical Research Centre and the NIHR Blood and Transplant Research Unit. M.R.C. was also supported by a Medical Research Council New Investigator Research Grant MR/N024907/1, Chan-Zuckerberg Initiative Human Cell Atlas Technology Development Grant, Versus Arthritis Cure Challenge Research Grant (21777) and NIHR Research Professorship RP-2017-08-ST2-002. We thank Drs Carl Anderson and Velislava Petrova (Wellcome Sanger Institute, UK) for support and useful discussions around BCR analysis. This work was performed, in part, using resources provided by the Cambridge Service for Data Driven Discovery (CSD3) operated by the University of Cambridge Research Computing Service (www.csd3.cam.ac.uk), provided by Dell EMC and Intel using Tier-2 funding from the Engineering and Physical Sciences Research Council (capital grant EP/P020259/1), and DiRAC funding from the Science and Technology Facilities Council (www.dirac.ac.uk).

## Author contributions

Conceptualization: O.S., M.R.C. Data curation: O.S., J.R.F. Formal analysis: O.S., M.R.C., Z.K.T., R.B.R. Methodology: O.S., M.R.C., J.R.F., R.B.R. Investigation: O.S., J.R.F., S.W., A.C., A.K.C., L.A., S.C., K.H., C.J.W. Resources: T.L., R.M., K.O., D.M. Funding acquisition: M.R.C., O.S. Project administration: M.R.C., O.S. Supervision: M.R.C., D.M., K.O., R.M. Writing–original draft: M.R.C., O.S. Writing – review & editing: M.R.C., O.S., J.R.F., R.B.R.

## Competing interests

The authors declare no competing interests.
