## [Peer Review File · Nature Communications]

Tissue-resident B cells orchestrate macrophage polarisation and functionREVIEWER COMMENTS

Reviewer #1 (Remarks to the Author):

In this paper, the authors showed the presence of tissue-resident B cells (predominantly B-1a cells) in many tissues. The authors characterized these tissue-resident B cell behaviors against microbiome and bacterial challenges. In a renal tract infection model using UPEC, there was no significant change in extravascular B cell numbers but extravascular BCR diversity and unique clonotypes numbers became similar to the intravascular compartment. In the urinary tract infection model using μ MT (deficient B-2 and a low number of B-1a cells) and p110 δ mice (increased number of B-1a cells), the authors showed a reduced number of bacterial CFUS in the bladder and kidney of μ MT mice while more CFUs were found in p110 δ mice. The authors showed that B-1a cell number is correlated with macrophage polarization (M1 vs M2) and chemokine expression, which is regulated by IL-10, highly secreted by tissue-resident B-1a cells. With a low number of extravascular B-1a cells (with lower IL-10), MNPs became more M1 macrophages and showed more uptake of the bacteria, while with a high number of B-1a cells (with high IL-10), MNPs became more M2 macrophages and showed less uptake of bacteria. From these results, the authors concluded that tissue-resident B cells regulate macrophage polarization via IL-10.

The authors elegantly performed experiments to examine the role of tissue-resident B cells, especially using the renal tract infectious model. The results firmly support the author's conclusion.

It has not been recognized the presence of tissue-resident B cells (B-1a cells) except for peritoneal and pleural cavities, thus, it is of interest to understand their roles in these tissues such as lung, liver, kidney, and bladder.

However, one major concern is the significance of the results. Are the results correlated with the infectious disease outcomes in a physiological setting?

For example, are patients with SCID or agammaglobulinemia more resistant to urinary tract infection than healthy people? If the presence of tissue-resident B-1a cells increases M2 macrophages and makes the host susceptible to infection, what is the significance of these tissue-resident B-1a cells?

Is there any effect of antibody production of B-1a cells among the animal?

B-1a cells are reported as progenitors of immune response activator B cells in the peritoneal cavity and lung (Science 2012, JEM 2014), protecting hosts from bacterial infections by secreting GM-CSF and polyclonal IgM. Are there any such protection roles of extravascular B-1a cells in the bladder and kidney?

How about the later outcomes of μ MT and p110 δ mice infected with IPEC? In the method, it is written that "mice were sacrificed at a specified time point (6hr, 12 hr., or 6 weeks)". The figures show only 6 hr results.

This work is beautifully done, but this reviewer is not sure about the advantage of this work in the field.

Reviewer #2 (Remarks to the Author):

Suchanek, Clatworthy et al study B cells in non-traditional B cell reservoirs. They perform a tour de force studying many different strains of mice including germ free mice, parabiotic experiments, human studies, RNA seq, pet store mice, an E. Coli infection model, sequencing studies and elegant macrophage polarization work.

Comments

1. The approaches are rigorous and largely descriptive, and give a broad picture of B cells which will be useful to the immunology community. I am used to more detailed work diving vertically into

a specific hypothesis rather than spreading outwards, but given the wealth of new information I feel that this work will be very important and highly cited.

2. Method section the pet store mouse work was performed in Minnesota but I do not see any Minnesota affiliations in the author list.

3. Kidney resident B cells have been studied, and implicated in repair after acute kidney injury. These processes likely involve tubular epithelial cells and macrophages, and can be discussed (Jang HR et al J Am Soc Nephrol 2010) PMID 20203156

4. If there are no germs in the germ free mice, what is the stimulus for B cells to reside in peripheral organs? The authors have studied details of the B cells in the E. Coli exposed mice. Would also be interesting if they went in depth for comparison in the germ free mice.

Reviewer #3 (Remarks to the Author):

This manuscript shows that tissue resident memory B cells (BRM) are found in multiple non-lymphoid tissues, via exclusion of circulating cells with antibody. Many of the BRM cells are B1a cells. These data were confirmed using parabiosis. BRM cells in some (but not all) tissues can be depleted with anti-CD20, perhaps reflecting local access to ADCC mechanisms. The frequency of BRM cells is variable between strains, and is decreased in GF mice and increased in mice co-housed with pet-shop mice. The BCR repertoire of BRM cells in the kidney is distinct from that in the blood, but infection tends to normalize the differences, perhaps to the recruitment of a more diverse repertoire. The absence of B cells (everywhere) makes the outcome of bladder infection better (less CFU), whereas an increase in B1a-like cells (everywhere) makes the outcome worse (more CFU). Myeloid cells in B cell replete mice are worse at phagocytosis, whereas those in B cell deficient mice are better. An increase in B1a-like cells worsens the ability of myeloid cells to phagocytose. B1a cells are a potent source of IL-10, whereas the isotype switched cells and the naïve B cells do not make IL-10. Using B cell-specific IL-10-deficient mice, they showed that monocytes are more M1-like and less M2-like. However, they do not show that B cell-specific IL-10-deficient mice are functionally improved in phagocytosis or bacterial killing.

The descriptive survey of BRM cells in various tissues, from various mice and under GF, SPF or pet-shop conditions is nice. The data showing a role for B cells in bacterial clearance recapitulates their earlier study in Nat Comm. It makes sense that BRM (rather than circulating B cells) would be responsible for altering monocyte function, but there is no data showing that tissue-resident cells are the ones responsible. Perhaps monocytes are programmed by B cells in the BM before they are recruited to the tissues??

Another possibility is antibody. Natural antibodies (made by B1 cells) are important for responses to bacteria and antibodies of all kinds are absent from uMT mice. Conversely, the antibody repertoire is undoubtedly impacted in the transgenic mice with increased B1a compartment. Can the authors distinguish the effects of B cells, from the effects of antibodies? Antibody transfers? Mice with B cells that cannot secrete antibody?

The idea that tissue-resident B1a cells make IL-10 and polarize macrophages in a non-functional way is important, but poorly tested. Using the B cell-specific IL-10 deficient mice, they show that gene expression in monocytes is altered, but they do not perform any functional assays. Do these mice clear bacteria better? Do monocytes from these mice exhibit increased phagocytic function?

I think IL-10 produced by B1a B cells (perhaps in tissues, perhaps not) does impact gene expression in monocytes, but it is not at all clear whether this change is the cause of altered bacterial clearance.

RESPONSE TO REVIEWER COMMENTS

Reviewer #1 (Remarks to the Author):

1. One major concern is the significance of the results. Are the results correlated with the infectious disease outcomes in a physiological setting? For example, are patients with SCID or agammaglobulinemia more resistant to urinary tract infection than healthy people? If the presence of tissue-resident B-1a cells increases M2 macrophages and makes the host susceptible to infection, what is the significance of these tissue-resident B-1a cells?

Whilst the impact of these human immunodeficiency syndromes on tissue B cells is an interesting question, the susceptibility of these patients to urinary tract infection is unlikely to be specifically related to defects in tissue B cells. SCID patients are deficient in all T and B cells, representing a profound and global immune defect with wide reaching effects on systemic and tissue immunity. Similarly, the comparison between X-linked agammaglobulinaemia and μ MT⁻ mice is problematic because Btk is not only essential for B-cell development but also for macrophage pro-inflammatory polarization¹. These confounding factors limit the extent to which the clinical phenotype of SCID or agammaglobulinemic patients can provide information on the potential protective effect of missing IL10-secreting tissue-resident B cells on organ susceptibility to infection.

In terms of clinical significance of our study, whilst there is controversy about the existence of B-1 cells in humans, our recent work provides strong evidence for their presence in developing human organs, including the kidney². There is also an emerging body of evidence suggesting the presence of innate-like B cells in human non-lymphoid organs (NLOs)^{2,3,4}, which may well share functional similarities with the tissue-resident mouse B-1 cells examined in our work and their importance may well extend beyond anti-bacterial defence; Tissue-resident B cell-derived IL10 may also affect macrophage function in tissue remodelling and repair⁵ or influence anti-cancer responses. Indeed, Wong et al. showed that intra-tumoral injection of B-1 cells lead to anti-inflammatory polarization of tumour-associated macrophages⁶.

2. Is there any effect of antibody production of B-1a cells among the animal? B-1a cells are reported as progenitors of immune response activator B cells in the peritoneal cavity and lung (Science 2012, JEM 2014), protecting hosts from bacterial infections by secreting GM-CSF and polyclonal IgM. Are there any such protection roles of extravascular B-1a cells in the bladder and kidney?

We agree with the reviewer that one might expect tissue B cells to have a protective role via antibody production, but our experiments show that this is not the case. Indeed, when designing our initial UTI experiments, we hypothesized that mice deficient in B cells would be more susceptible to infection and those with increased numbers of B cells would be protected, presumably by natural antibody secretion. Surprisingly, our experiments showed the opposite result. Hence, we focused the current study on exploring the mechanism by which tissue-resident B cells drive this organ susceptibility to infection, which was unlikely to be antibody-related.

3. How about the later outcomes of μ MT and p110 δ mice infected with UPEC? In the method, it is written that "mice were sacrificed at a specified time point (6hr, 12 hr., or 6 weeks)". The figures show only 6 hr results.

The specified time points of mice sacrifice following UPEC inoculation described in Methods covers all UTI experiments performed throughout this manuscript. 6-week time point was used only in the BCR-seq experiment, examining the perturbation of kidney B-cell repertoire following a resolved pyelonephritis (Fig. 4). UTI experiments with μ MT⁻ and PI3K KI mice had only early time points (6-12 hrs) to exclude the confounding effects of adaptive (B-2) immune response.

Indeed, Stark *et al.* showed that although μMT^- mice had a significantly lower early mortality from pneumonia compared to WT, at day 30, surviving μMT^- mice had a much lower bacterial clearance in their lungs compared to WT mice surviving to this time point. This suggests that at early time points B cells (B-1 cells) increase susceptibility to infection while at later stages (B-2 cells) might be required for adaptive / antibody-dependent bacterial clearance⁷. Our study focused on the former.

4. This work is beautifully done, but this reviewer is not sure about the advantage of this work in the field.

We thank the reviewer for their comments, but would respectfully note that our work highlights several previously unappreciated facets of B cell biology; Our work has shown that innate-like B(-1) cells seed into NLOs together with macrophages and orchestrate their homeostatic anti-inflammatory polarization, with important implications for future studies of tissue immunity and its potential therapeutic modulation by B cell depleting agents. To facilitate the communication of this key message to other researchers / clinicians we added the following graphical abstract in the revised manuscript as **Figure 8**.

Reviewer #2 (Remarks to the Author):

1. Method section the pet store mouse work was performed in Minnesota, but I do not see any Minnesota affiliations in the author list.

The pet store animal work and parabiosis was indeed performed at the University of Minnesota by our collaborators Sathi Wijeyesinghe and David Masopust (PI) who are both on the authors list with their appropriate affiliation.

2. Kidney resident B cells have been studied, and implicated in repair after acute kidney injury. These processes likely involve tubular epithelial cells and macrophages, and can be discussed (Jang HR et al J Am Soc Nephrol 2010) PMID 20203156.

The study by Jang *et al.* observed an influx of B-1 cells into kidneys following an ischaemic reperfusion injury (IRI) driven by increased CXCL13 expression. Kidney B-1 and plasma cells were associated with slower tissue regeneration, increased tubular atrophy/fibrosis and ultimately poorer kidney function. Our group and others have also described an influx of B-2 cells into the kidney in the context of AKI⁸.

However, the novelty and importance of the current study, is the identification of tissue-resident B cells, including B-1 cells, in homeostasis (i.e. in the absence of injury), not only in the kidney but also other non-lymphoid organs. We demonstrate IL10-mediated anti-inflammatory effects of kidney B cells, consistent with the anti-inflammatory / pro-fibrotic effect of kidney B-1 cells after injury described by Jang *et al.*, but they did not identify the underlying mechanism behind this regulation, in contrast to our study.

Given the reviewers suggestion, we have now referenced this study in the manuscript Introduction, as follows:

“Although B-1 cells occupy body cavities in homeostasis, their presence in skin, lung and kidney has been described in the context of injury or infection.”

3. If there are no germs in the germ-free mice, what is the stimulus for B cells to reside in peripheral organs? The authors have studied details of the B cells in the E. Coli exposed mice. Would also be interesting if they went in depth for comparison in the germ-free mice.

The presence of tissue B cells in GF mice is interesting. In response to the reviewer’s comment, we have now added a panel assessing B-1a cell abundance in the extravascular B cell compartment in GF in comparison with SPF mouse organs (**revised Fig. 3b**).

Figure 3b (revised)

The reviewer is correct, and we also concluded, that the presence of tissue B cells in germfree mice suggests that neonatal seeding of B cells to tissues is microbiome independent, as is the case for macrophage tissue seeding. The stimulus for immune cell recruitment to organs is the expression of local chemokines. Our experiments show that for B cells, this does not require microbial colonisation. The fact that B cells are present in neonatal kidneys is consistent with the conclusion that chemokine expression is part of the normal organ development. Indeed, we performed an assessment of human neonatal kidney single cell transcriptomes and found the expression of B-cell attracting chemokines (e.g. CXCL12) across a variety of different kidney cell types pre-natally (Fig. C3), supporting this conclusion. Therefore, although the microbiome is not required for initial colonisation, our data shows that it does affect the number of extravascular B cells in non-lymphoid organs (NLOs), which was significantly lower in germ-free mice.

Figure C3. Human and mouse neonatal kidneys express B-cell attracting chemokine CXCL12. (a) UMAP embedding plot based on transcriptome of single cells recovered from three compartments (stromal, immune and nephron/endothelial) of foetal human kidneys and annotated as per Stewart *et al.*⁹. (b) CXCL12 expression by cells displayed in (a). (c) CXCL12 mean expression dot plots for all cell subsets from the stromal (left) and nephron/endothelial compartment (right). (d) CXCR4 (i.e. CXCL12 receptor) mean expression dot plot for immune cells found in (a). (e) UMAP embedding plot based on transcriptome of single cells recovered from neonatal mouse kidney and annotated as per Miao *et al.*¹⁰ (f) Cxcl12 mean expression dot plot for cell subsets displayed in (e). (g) Focused UMAP embedding plot of immune cells showed in (e) with a mean expression dotplot of B cell/plasma cell marker genes and Cxcr4. Circle size corresponds to fraction of cells expressing the gene and colour gradient corresponds to relative mean expression scaled from 0 to 1.

Reviewer #3 (Remarks to the Author):

1. The descriptive survey of BRM cells in various tissues, from various mice and under GF, SPF or pet-shop conditions is nice. The data showing a role for B cells in bacterial clearance recapitulates their earlier study in Nat Comm. It makes sense that BRM (rather than circulating B cells) would be responsible for altering monocyte function, but there is no data showing that tissue-resident cells are the ones responsible. Perhaps monocytes are programmed by B cells in the BM before they are recruited to the tissues?

Delineating the local tissue effects of B cells versus systemic effects, including in bone marrow is difficult, and we are not aware of any methodology currently available that would allow us to do this. That said, our data does convincingly show an effect of B cell derived IL10 (using *Cd19^{cre}//10^{fl/fl}* mouse model), on the polarization of tissue-resident macrophages (not only monocytes) both in kidneys and bladders (**revised Fig. 7f-h, revised Fig. S8b-c**). This includes F4/80^{hi} macrophages that have previously been fate-mapped to demonstrate that this subset is predominantly yolk-sac derived, and in our study, is transcriptionally similar to these fate-mapped yolk-sac derived macrophages. This is consistent with the conclusion that B cell derived IL10 affects macrophages that are not monocyte derived (excluding bone marrow effects), and are long term residents in the kidney and bladder, strongly arguing for a local effect within the tissue. We agree with Reviewer 3 that it is possible that the anti-inflammatory polarization of CD11b high monocyte-derived macrophages by B-1 cells might have occurred in the bone marrow before seeding into non-lymphoid organs (NLOs), although B-1 cells are rare within the bone marrow, making up <1% of B cells¹¹. In addition, several other findings we presented in this study altogether suggest that macrophage polarization takes place, at least in part, inside NLOs (i.e. under the local influence of tissue-resident B-1 cells):

1. B-1 cells are the main source of IL10 within the extravascular B cell pool in NLOs (**Fig. 7b, 7c, S7**);
2. Extravascular B-1 cells are tissue-resident (i.e. most sessile extravascular B-cell subset) (**Fig. 2b**);
3. Extravascular B-cells are spatially co-localized with tissue MNPs (**Fig. 6e**);
4. B-cell deficient μ MT kidneys contain lower number of *Il10* transcripts compared to WT kidneys (**Fig. 7a**);
5. B-1 cell rich PI3K KI kidneys contain higher number of *Il10* transcripts compared to WT kidneys (**Fig. 7a**);
6. Percentage of extravascular B-1 cells significantly correlates with the percentage of anti-inflammatory (CD206⁺) macrophages within each organ (**Fig. 6d**).

To discuss this issue, we added the following text into the revised manuscript (Discussion):

“Our finding that resident B-1 cells were an important cellular source of tissue IL10 both in WT and $p110\delta^{E1020K-B}$ and that the abundance of these cells significantly correlated with the proportion of anti-inflammatory (CD206⁺) macrophages within each organ is consistent with the conclusion that this B cell-mediated macrophage polarization takes place, at least in part, locally.”

2. Another possibility is antibody. Natural antibodies (made by B1 cells) are important for responses to bacteria and antibodies of all kinds are absent from μ MT mice. Conversely, the antibody repertoire is undoubtedly impacted in the transgenic mice with increased B1a compartment. Can the authors distinguish the effects of B cells, from the effects of antibodies? Antibody transfers? Mice with B cells that cannot secrete antibody?

We agree with Reviewer 3 that IL-10 secretion might not be the only mediator of B-1 cell driven anti-inflammatory macrophage polarization in NLOs.

There is a large body of evidence supporting the protective role of natural antibodies against a broad spectrum of infections¹². Hence, we were surprised by our finding of the superior bacterial clearance and pro-inflammatory macrophage polarization in B-cell deficient μ MT mice (lacking natural antibodies), and the opposite in PI3K KI mice. If natural antibodies were responsible for this phenomenon, their anti-inflammatory effect on nearby tissue MNPs would have to outweigh their protective role, which would be not in line with

previous studies¹². Moreover, there is currently no evidence for the expression of inhibitory Fc receptor on mouse MNPs that would bind IgM, the most abundant natural antibody isotype¹³. (There is, however, a possibility that some MNP inhibition could be mediated by the less abundant natural monomeric IgA via FcαRI¹⁴).

As mentioned in the previous comment, to overcome the limitations of the μ MT⁻ and PI3K KI mouse models where B / B-1 cells numbers are altered, we profiled macrophage polarization / function and bacterial tissue clearance also in *Cd19^{cre}Il10^{fl/fl}* mouse model as part of this manuscript revision. We reflected this comment by adding the following text into the Discussion:

*“One caveat is that the two genetically modified mouse models we used (μ MT⁻ and *p110 δ ^{E1020K-B}* mice) differ from controls not only in their number of IL10-producing tissue-resident B cells but also, for example, in their B-cell signaling, selection and antibody repertoire, which may modulate infection susceptibility^{7, 15}. However, we found pro-inflammatory transcriptomic and functional changes in tissue macrophage, as well as increased bacterial clearance, in B-IL10-KO mice, mirroring our observations from μ MT⁻ animals, confirming the importance of B-cell derived IL10 in shaping macrophage polarization.”*

3. The idea that tissue-resident B1a cells make IL-10 and polarize macrophages in a non-functional way is important, but poorly tested. Using the B cell-specific IL-10 deficient mice, they show that gene expression in monocytes is altered, but they do not perform any functional assays. Do these mice clear bacteria better? Do monocytes from these mice exhibit increased phagocytic function?

We agree with the reviewer that demonstrating the impact of B-cell derived IL10 on tissue macrophage function, in addition to presented (pro-inflammatory) transcriptomic changes, will make our study more powerful. We did not perform UTI or phagocytosis experiments in *Cd19^{cre}Il10^{fl/fl}* mice before the initial manuscript submission because these animals were not available in our animal facility.

In response to the reviewer’s suggestion, we re-derived these mice into our animal facility, requiring an MTA and the challenges of establishing a working colony. This inevitably introduced a significant delay into the rebuttal process. However, we have now performed an *ex vivo* phagocytosis assay, using kidney macrophages from *Cd19^{cre}Il10^{fl/fl}* mice and controls, and demonstrating enhanced phagocytosis in macrophages obtained from mice with B cell intrinsic IL10 deficiency. These data are shown below and included in the revised manuscript as follows:

“To test whether these transcriptomic changes have also an impact on kidney macrophage function, we performed an ex-vivo phagocytosis assay with fluorescently labelled E. coli bioparticles. We found a significantly higher phagocytic ability of both macrophage subsets from B-IL10-KO kidneys when compared to controls (Fig. 7h).”

Figure 7 (revised)

4. I think IL-10 produced by B1a B cells (perhaps in tissues, perhaps not) does impact gene expression in monocytes, but it is not at all clear whether this change is the cause of altered bacterial clearance.

To address this point further, we performed an additional set of UTI experiments (cystitis model) in B-IL10-KO mice and found improved UPEC clearance in B-IL10-KO mice compared to controls, accompanied by a higher neutrophil and monocyte recruitment in the tissue (Shown below and in revised Fig. S8b-c).

We have updated the manuscript accordingly:

*Finally, to further investigate the effect of IL10 on tissue MNP's polarisation at steady state, we treated WT mice with an IL10 neutralising antibody for one week and assessed bladder MNP polarization (Fig. S8B). We found a significant reduction in the proportion of CD206-expressing MNPs, and higher expression of pro-inflammatory gene *Tnfa* in mice treated with the anti-IL10 antibody compared with isotype control (Fig. S8B).*

“Finally, to test the functional importance of B-cell derived IL10 on local bacterial clearance in vivo, we performed a UTI (cystitis model) in B-IL10-KO mice. We observed significantly greater UPEC clearance from the bladder in B-IL-10 KO mouse compared to controls (Fig. S8b), accompanied by a higher neutrophil and monocyte recruitment to the tissue (Fig. S8c).”

Figure S8 (revised)

References:

1. Good, L., Benner, B. & Carson, W.E. Bruton's tyrosine kinase: an emerging targeted therapy in myeloid cells within the tumor microenvironment. *Cancer Immunol Immunother* **70**, 2439-2451 (2021).
2. Suo, C. *et al.* Mapping the developing human immune system across organs. *Science (New York, N.Y.)* **376**, eabo0510 (2022).
3. Geherin, S.A. *et al.* IL-10+ Innate-like B Cells Are Part of the Skin Immune System and Require $\alpha 4\beta 1$ Integrin To Migrate between the Peritoneum and Inflamed Skin. *Journal of immunology (Baltimore, Md. : 1950)* **196**, 2514-2525 (2016).
4. Asano, Y. *et al.* Innate-like self-reactive B cells infiltrate human renal allografts during transplant rejection. *Nat Commun* **12**, 4372 (2021).
5. Mantovani, A., Biswas, S.K., Galdiero, M.R., Sica, A. & Locati, M. Macrophage plasticity and polarization in tissue repair and remodelling. *J Pathol* **229**, 176-185 (2013).
6. Wong, S.C. *et al.* Macrophage polarization to a unique phenotype driven by B cells. *European journal of immunology* **40**, 2296-2307 (2010).
7. Stark, A.-K. *et al.* PI3K δ hyper-activation promotes development of B cells that exacerbate Streptococcus pneumoniae infection in an antibody-independent manner. *Nature Communications* **9**, 3174 (2018).
8. Inaba, A. *et al.* B Lymphocyte-Derived CCL7 Augments Neutrophil and Monocyte Recruitment, Exacerbating Acute Kidney Injury. *Journal of immunology (Baltimore, Md. : 1950)* **205**, 1376-1384 (2020).
9. Stewart, B.J. *et al.* Spatiotemporal immune zonation of the human kidney. *Science (New York, N.Y.)* **365**, 1461-1466 (2019).
10. Miao, Z. *et al.* Single cell regulatory landscape of the mouse kidney highlights cellular differentiation programs and disease targets. *Nature Communications* **12**, 2277 (2021).
11. Choi, Y.S., Dieter, J.A., Rothausler, K., Luo, Z. & Baumgarth, N. B-1 cells in the bone marrow are a significant source of natural IgM. *European journal of immunology* **42**, 120-129 (2012).
12. Smith, F.L. & Baumgarth, N. B-1 cell responses to infections. *Current opinion in immunology* **57**, 23-31 (2019).
13. Bruhns, P. & Jonsson, F. Mouse and human FcR effector functions. *Immunol Rev* **268**, 25-51 (2015).
14. Breedveld, A. & van Egmond, M. IgA and Fc α RI: Pathological Roles and Therapeutic Opportunities. *Front Immunol* **10**, 553 (2019).
15. Wray-Dutra, M.N. *et al.* Activated PI3KCD drives innate B cell expansion yet limits B cell-intrinsic immune responses. *The Journal of experimental medicine* **215**, 2485-2496 (2018).

REVIEWERS' COMMENTS

Reviewer #1 (Remarks to the Author):

There are no other comments.

Reviewer #2 (Remarks to the Author):

The authors have significantly improved the manuscript. The work is highly significant and will be an important basis for future studies.

Reviewer #3 (Remarks to the Author):

The authors have responded appropriately to my comments. Importantly, they now show that mice lacking IL-10 made by B cells more efficiently reduce bacterial burden in the bladder of infected mice. I am a bit surprised that they buried this data in Fig S8 rather than putting it in a main figure. The result seems to be a primary conclusion of the paper.

Suchanek *et al.* '*Tissue-resident B cells orchestrate macrophage polarisation and function*'. R2.
Ref NCOMMS-22-24929A.

Response to reviewers' comments:

Reviewer #1 (Remarks to the Author):

There are no other comments.

Reviewer #2 (Remarks to the Author):

The authors have significantly improved the manuscript. The work is highly significant and will be an important basis for future studies.

Reviewer #3 (Remarks to the Author):

The authors have responded appropriately to my comments. Importantly, they now show that mice lacking IL-10 made by B cells more efficiently reduce bacterial burden in the bladder of infected mice. I am a bit surprised that they buried this data in Fig S8 rather than putting it in a main figure. The result seems to be a primary conclusion of the paper.

Thank you for this useful comment. We agree that this is important data and have now included the panel showing reduced bacterial burden in urinary bladders of mice with IL10-deficient B cells in the main figure (Fig. 7i-j).